# Towards Hierarchy–Uniformity Equilibrium: Recovering Semantic Depth in Hypergraph Contrastive Learning

**Ruiting Zhao** [1]  **Ming Li** [2]  **Lixin Cui** [3]  **Lu Bai** [4]  **Feilong Cao** [5]  **Ke Lv** [6 7]  **Pietro Lio** [8]

## Abstract

Hypergraph contrastive learning is an effective paradigm for representation learning on higher-order relational data, yet existing methods largely ignore that hyperedges link nodes with multi-level semantics. Standard contrastive objectives emphasize instance discrimination via hyperspherical uniformity and tend to push embeddings apart in an indiscriminate manner. We show that this leads to a *Hierarchy—Uniformity Conflict*, whose geometric manifestation is *Semantic Flattening*, where the semantic depth of hyperedges collapses into a nearly flat cloud of instances. To address this issue, we introduce **HyperDepth**, a hypergraph contrastive learning framework that moves representations towards a hierarchy–uniformity equilibrium by jointly coordinating spectral and geometric signals. HyperDepth employs a decoupled spectral encoding scheme with adaptive gating so that high-frequency components focus on local instance discrimination while low-frequency components capture global hierarchical structure. On top of this, an energy-based hierarchical alignment module attaches a learnable prototype tree to the representation space and minimizes an interpretable energy functional to recover the semantic depth of hyperedges. Theoretically, under a mild frequency-separation assumption, we show that the local contrastive and global hierarchical objectives operate on orthogonal spectral components and admit equilibrium embeddings that preserve semantic depth while still retaining instance-level discrimination. Experiments on 15 hypergraph datasets and 17 supervised and self-supervised baselines, spanning homophilic and heterophilic regimes, show that HyperDepth attains strong performance with the best average rank.

## 1. Introduction

**Background.** Hypergraph learning provides a principled framework for modeling higher-order relational data by representing multi-way interactions among entities as hyperedges (Gao et al., 2020; Antelmi et al., 2023). This perspective has been adopted in various domains where group-wise or set-valued relations naturally arise, and there is a growing need to learn informative hypergraph representations from such data. As labeled hypergraph data are often scarce or expensive to obtain, self-supervised approaches for hypergraph representation learning (Wei et al., 2022; Lee & Shin, 2023; Song et al., 2024) have attracted increasing attention, with hypergraph contrastive learning emerging as a key paradigm.

**Motivation.** Hypergraph contrastive learning has become a central paradigm for self-supervised hypergraph representation learning, typically aligning augmented views of the same instance while pushing apart different instances. Yet existing methods largely overlook that real-world hyperedges encode not only whether a group of nodes is related, but also how these relations are organized across multiple semantic levels. In many settings, the nodes in a hyperedge share a hierarchical structure of interests or attributes rather than a single flat label. As illustrated in Figure 1 with a co-authorship example, a hyperedge connecting a group of authors may reflect a nested semantic organization, from a broad research area (e.g., *Deep Learning*), to a more specific domain (e.g., *GNN*), and further down to a fine-grained focus (e.g., *theory versus application*). Ideally, node representations should preserve such *semantic depth*, so that authors related at different levels remain organized along a hierarchy in the embedding space. In contrast, preva-

[1]School of Computer Science and Technology, Zhejiang Normal University, Jinhua, China [2]Zhejiang Key Laboratory of Intelligent Education Technology and Application, Zhejiang Normal University, Jinhua, China [3]Central University of Finance and Economics, Beijing, China [4]School of Artificial Intelligence, Beijing Normal University, Beijing, China [5]School of Mathematical Sciences, Zhejiang Normal University, Jinhua, China [6]School of Engineering Science, University of Chinese Academy of Sciences, Beijing, China [7]Peng Cheng Laboratory, Shenzhen, China [8]Department of Computer Science and Technology, University of Cambridge, UK. Correspondence to: Ming Li <mingli@zjnu.edu.cn>.

*Proceedings of the 43rd International Conference on Machine Learning*, Seoul, South Korea. PMLR 306, 2026. Copyright 2026 by the author(s).

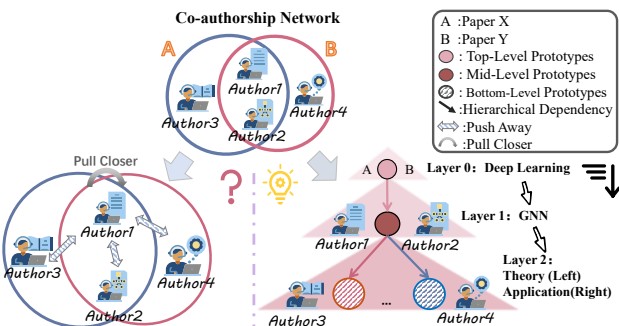

*Figure 1.* Illustration of the hierarchy–uniformity conflict. **(Left):** Indiscriminate hyperspherical uniformity in hypergraph contrastive learning causes *semantic flattening*, pushing semantically related nodes apart. **(Right):** HyperDepth mitigates this by aligning embeddings to a hierarchical prototype tree, preserving semantic depth while retaining discriminability.

lent hypergraph contrastive objectives emphasize instance discrimination through hyperspherical uniformity: embeddings are encouraged to spread out as evenly as possible on a unit sphere. This uniformity-driven objective applies an essentially indiscriminate "push-away" force that treats co-authors merely as distinct instances, regardless of their higher-level semantics.

We refer to this tension between preserving hierarchical relations and enforcing uniform spreading as the **Hierarchy–Uniformity Conflict**. Its geometric consequence is what we term **Semantic Flattening**: the multi-granular hierarchy encoded by hyperedges collapses into an almost single-layer cloud of nearly equidistant points. Existing hypergraph contrastive methods primarily optimize for uniformity and instance-level discrimination, without an explicit mechanism to maintain or recover the semantic depth of hyperedges. This raises a key question:

> *"How can we retain the benefits of contrastive discrimination while preventing the underlying **hyperedge hierarchies** from being flattened?"*

**Method.** We address this question by proposing **Hyper-Depth**, a hypergraph contrastive learning framework that steers representations towards a *hierarchy–uniformity equilibrium*. HyperDepth is built on the observation that local instance uniqueness and global hierarchical structure are mainly carried by different spectral components: high-frequency variations capture fine-grained distinctions between nearby nodes, whereas low-frequency components encode smooth patterns consistent with coarse semantic groupings. Accordingly, HyperDepth employs a decoupled spectral encoder with adaptive gating that separates and recombines low- and high-frequency signals, encouraging the high-frequency channel to support node-level contrastive discrimination while the low-frequency channel preserves hierarchical coherence. On top of this encoder, an energy-

based hierarchical alignment module attaches a learnable prototype tree to the representation space and minimizes a geometric energy functional that anchors each node to appropriate prototypes along a path, recovering the semantic depth that standard uniformity objectives tend to flatten. In this way, hierarchical organization and hyperspherical uniformity are enforced in complementary parts of the embedding space rather than being in direct conflict. We further validate HyperDepth through both theoretical analysis (with detailed proofs in **Appendix B**) and extensive experiments.

**Contributions.** This work makes three main contributions.

- We propose HyperDepth, a hypergraph contrastive learning framework that explicitly targets the Hierarchy–Uniformity Conflict by combining a decoupled spectral encoder with adaptive gating, an energy-based hierarchical alignment module, and a local contrastive branch defined in a projection space.

- We provide a geometric formulation of the tension between instance-level uniformity and hierarchical semantics, introduce the notion of a hierarchy–uniformity equilibrium, and prove theoretically that under a mild frequency-separation assumption the local and global objectives act on orthogonal spectral components and admit equilibrium embeddings that preserve both semantic depth and discriminability.

- We conduct experiments on 15 benchmark datasets, including both homophilic and heterophilic settings, against 17 supervised and self-supervised baselines, and show that HyperDepth achieves strong overall performance with the best average rank.

## 2. Related Work

### 2.1. Hypergraph Neural Networks

Hypergraph neural networks (HNNs) provide a principled framework for modeling higher-order relations beyond pairwise graphs. Early methods mostly adapt spectral graph convolution or structural expansion to the hypergraph setting. HGNN (Feng et al., 2019) and HGNN$^+$ (Gao et al., 2023) extend graph convolutions via spectral approximation and spatial fusion, while HyperGCN (Yadati et al., 2019) approximates hyperedges by pairs of nodes connected through mediators to reduce computation. To avoid the distortions introduced by clique or star expansions, set-based architectures such as AllDeepSets and AllSetTransformer (Chien et al., 2022) treat each hyperedge as an unordered set and aggregate features with permutation-invariant operators, enabling flexible handling of variable-size hyperedges.

Recent work further enriches HNN architectures with equivariance, dynamics, and attention. ED-HNN (Wang

et al., 2023a), HyperND (Prokopchik et al., 2022), and HDS$^{\text{ODE}}$ (Yan et al., 2024) incorporate equivariant constraints, nonlinear diffusion, and neural ODE formulations to model hypergraph signals with richer continuous-time dynamics. HGNNv2 (Gao et al., 2026) proposes a hypergraph dynamical system based on partial differential equations, introducing a position-aware anisotropic diffusion term and an external control term to stabilize HNN training. HyperGT (Liu et al., 2024) injects hypergraph structure into Transformer backbones, combining higher-order connectivity with attention-based sequence modeling. From a spectral perspective, however, many of these models still behave as predominantly low-pass filters and are mainly designed for supervised learning with sufficient labels. SensHNN (Yadati, 2025) shows that standard message passing can suffer from oversquashing on hypergraphs, and FrameHGNN (Li et al., 2025a) analyzes hypergraph frequency responses and introduces sheaflet-based high-pass components to alleviate oversmoothing. Yet these architectures are not explicitly tailored to self-supervised objectives, and the question of how to exploit multi-frequency behavior in a purely unsupervised setting, while distinguishing high-frequency semantic signals from noise, remains largely open.

## 2.2. Self-Supervised Hypergraph Learning

Self-supervised learning (SSL) has become an important paradigm for hypergraph representation learning (Sun et al., 2023; Feng et al., 2025), aiming to leverage structural and semantic regularities without labels. Existing approaches roughly fall into generative and contrastive families. Generative methods such as HypeBoy (Kim et al., 2024) learn representations by modeling hyperedge completion or reconstruction, providing a probabilistic view of higher-order relations.

The dominant line of work, however, is contrastive hypergraph learning. HyperGRL (Du et al., 2022) introduces dual-level contrastive objectives between nodes and hyperedges to jointly shape their embeddings. HyperGCL (Wei et al., 2022) employs learnable augmentation generators to construct diverse hypergraph views for contrastive training. TriCL (Lee & Shin, 2023) performs tri-level alignment among nodes, hyperedges, and hyperedge groups to capture multi-granular structure. S2-HHGR (Zhang et al., 2021) and related methods (Xia et al., 2022) couple graph and hypergraph objectives to mitigate sparsity in recommendation scenarios. HyFi (Roh et al., 2024) reduces the cost of heavy structural perturbations by introducing weak positive samples for fine-grained discrimination, and MMACL (Lee & Chae, 2024) designs a mixed-attention mechanism to fuse pairwise and higher-order dependencies across multiple views. HyperAim (Li et al., 2026) proposes a contrastive framework that integrates adaptive multi-frequency filtering through decoupled and coupled designs to better exploit

spectral information in hypergraphs.

## 2.3. Positioning of Our Work

The above HNN and SSL works highlight two trends that are directly relevant to our setting: (i) advanced hypergraph architectures increasingly expose multi-frequency behavior and higher-order structure, and (ii) contrastive objectives are the prevailing tool for self-supervised hypergraph learning. However, existing contrastive methods mainly emphasize instance-level discrimination and hyperspherical uniformity, while prior spectral studies focus on supervised training and do not explicitly consider the hierarchical semantics encoded by hyperedges. HyperDepth is positioned at this intersection: it starts from a geometric formulation of the hierarchy–uniformity conflict in hypergraph contrastive learning, and then designs a decoupled spectral encoder together with an energy-based hierarchical prototype module to move representations towards a hierarchy–uniformity equilibrium. In this way, HyperDepth builds on and complements prior architectures and contrastive frameworks, but explicitly targets the recovery of semantic depth and the control of spectral bias in a purely self-supervised setting.

## 3. Preliminaries

**Basics on Hypergraphs.** Let $\mathcal{H} = (\mathcal{V}, \mathcal{E})$ be a hypergraph with node set $\mathcal{V}$, hyperedge set $\mathcal{E}$, $|\mathcal{V}| = N$ nodes, and $|\mathcal{E}| = E$ hyperedges. Its structure can be represented by the incidence matrix $\mathbf{H} \in \{0,1\}^{N \times E}$, where $\mathbf{H}_{i,e} = 1$ if node $i$ belongs to hyperedge $e$ and $\mathbf{H}_{i,e} = 0$ otherwise. To analyze spectral properties, we consider the normalized hypergraph Laplacian

$$\mathbf{L} = \mathbf{I} - \mathbf{D}_v^{-1/2} \mathbf{H} \mathbf{D}_e^{-1} \mathbf{H}^\top \mathbf{D}_v^{-1/2},$$

where $\mathbf{D}_v$ and $\mathbf{D}_e$ are diagonal matrices of node degrees and hyperedge degrees, respectively.

**Contrastive Learning and Uniformity.** We employ the InfoNCE objective to learn discriminative representations. As shown in (Wang & Isola, 2020), as the number of negative samples $M \to \infty$, minimizing the InfoNCE loss is asymptotically equivalent to optimizing the following two terms:

$$\lim_{M \to \infty} (\mathcal{L}_{\text{NCE}} - \log M) = \underbrace{-\frac{1}{\tau} \mathbb{E}_{(i,j) \sim p_{\text{pos}}}[h_i^\top h_j]}_{\text{Alignment}}$$

$$+ \underbrace{\mathbb{E}_{i \sim p_{\text{data}}} \left[ \log \mathbb{E}_{q \sim p_{\text{data}}} \left[ e^{h_q^\top h_i / \tau} \right] \right]}_{\text{Uniformity}},$$

where $h_i$ denotes the representation of instance $i$ and $\tau$ is the temperature. The Alignment term encourages representations of positive pairs to be similar, while the Uniformity term encourages representations to spread out on the hypersphere. By penalizing regions of high density through the

log-expectation of exponential similarities, the Uniformity term acts as a soft repulsive force that mitigates representation collapse.

# 4. Theoretical Formulation and Geometric View of Hierarchy–Uniformity Conflict

To move beyond purely empirical observations, we provide a geometric formulation of the Hierarchy–Uniformity Conflict in hypergraph contrastive learning. We first introduce two structural objectives, local-discriminative and global-semantic, and then show how they induce incompatible distance preferences for semantically related nodes.

**Local-Discriminative Structure.** The local-discriminative structure requires the learned representation space to be informative enough to distinguish individual instances. Let $\mathbf{E} \in \mathbb{R}^{N \times d}$ denote the node embedding matrix generated by an encoder $f_\Phi$, and let $h_i \in \mathbb{R}^d$ denote the $i$-th row, normalized to the **unit hypersphere** ($\|h_i\|_2 = 1$). Maximizing discriminative capacity can be approximated by encouraging embeddings to spread uniformly on the unit sphere. A common proxy objective is to minimize the average Gaussian potential (RBF kernel) between arbitrary node pairs:

$$\min_{\mathbf{E}} \mathcal{L}_{\text{Local}} = \mathbb{E}_{i,j \sim \mathcal{V}} \left[ e^{-2\|h_i - h_j\|_2^2} \right]. \quad (1)$$

**Global-Semantic Structure.** The global-semantic structure captures the intrinsic hierarchical organization encoded by containment relations in hyperedges. Geometrically, this can be expressed as a form of hierarchical alignment on the hypersphere. Instead of requiring embeddings to be uniformly dispersed, we require them to be directionally aligned with a discrete set of hierarchical prototypes $\mathcal{C}$ that form a tree-structured hierarchy. Let $\mathcal{T}$ denote the set of valid paths in this prototype tree and $c_{\mathcal{P}}^{\text{leaf}}$ the leaf prototype associated with path $\mathcal{P} \in \mathcal{T}$. We then consider the following spherical concentration objective:

$$\max_{\mathbf{E},\mathcal{C}} \mathcal{J}_{\text{Global}} = \sum_{i \in \mathcal{V}} \max_{\mathcal{P} \in \mathcal{T}} \langle h_i, c_{\mathcal{P}}^{\text{leaf}} \rangle, \quad (2)$$

where $\langle \cdot, \cdot \rangle$ denotes cosine similarity.

**Definition 1 (Hierarchy–Uniformity Conflict).** *Let $\mathcal{H}$ be a hypergraph endowed with a semantic hierarchy, and let $\mathcal{S}_i$ denote the set of nodes that share a semantic ancestor with node $i$. Consider the local-discriminative objective $\mathcal{L}_{\text{Local}}$ and the global-semantic objective $\mathcal{J}_{\text{Global}}$ defined above. We say that a representation space exhibits a* Hierarchy–Uniformity Conflict *if, for some node $i$ with $\mathcal{S}_i \neq \emptyset$, the preferred pairwise distances for nodes in $\mathcal{S}_i$ under the two objectives lie in disjoint regimes, for all $j \in \mathcal{S}_i$,*

$$\underbrace{\|h_i - h_j\|_2 \xrightarrow{\mathcal{L}_{\text{Local}}} \sqrt{2}}_{\text{push away (uniformity)}} \quad vs. \quad \underbrace{\|h_i - h_j\|_2 \xrightarrow{\mathcal{J}_{\text{Global}}} 0}_{\text{pull closer (hierarchy)}}$$

*In other words, representations that nearly minimize $\mathcal{L}_{\text{Local}}$ push semantically related nodes towards near-orthogonality*

on the unit sphere, whereas those that nearly maximize $\mathcal{J}_{\text{Global}}$ *pull them towards coincidence, inducing incompatible geometric preferences for the same nodes.*

**Semantic Flattening.** When training is dominated by the local-discriminative objective, the repulsive force induced by uniformity applies to all node pairs, including those that share hierarchical semantics. As a consequence, intra-group distances for nodes in $\mathcal{S}_i$ are driven towards the same large distance regime as inter-group distances. In the embedding space, the nested structure encoded by hyperedges is no longer distinguishable: nodes that should occupy different levels of a hierarchy become arranged in an almost equidistant, nearly flat configuration on the hypersphere. We refer to this loss of semantic depth as *Semantic Flattening*.

**Goal of this work.** In this work, we explicitly take the Hierarchy–Uniformity Conflict in Definition 1 as the central modeling target. Our aim is to design a hypergraph contrastive learning framework that reconciles the two competing forces induced by $\mathcal{L}_{\text{Local}}$ and $\mathcal{J}_{\text{Global}}$. Concretely, we seek a representation space in which nodes remain locally distinguishable, preserving their individual identities, while simultaneously retaining semantic depth by aligning embeddings with hierarchical prototypes. From a spectral perspective, this amounts to allocating high-frequency components to local discrimination and low-frequency components to global hierarchical organization. The proposed framework, **HyperDepth**, detailed in the following section, is built precisely to implement this principle and to steer learning towards a *hierarchy–uniformity equilibrium*, rather than allowing uniformity to erase hierarchical structure.

# 5. Proposed Method

## 5.1. Framework Overview

We propose HyperDepth, a new hypergraph contrastive learning framework via jointly modeling spectral and geometric structure. As illustrated in Figure 2, given an input hypergraph and its augmented view, HyperDepth first applies a **Decoupled Spectral Encoder** that passes node features through complementary high-pass and low-pass filters, with shared weights and a learnable gating mechanism to adaptively fuse the two branches into spectrally enriched embeddings that retain both local variations and global trends. On top of these embeddings, the **Global Hierarchical alignment** module attaches a learnable prototype tree and uses an energy-based objective to align each node with a path in the hierarchy, encouraging semantically related nodes to form level-specific clusters while preserving their nested structure. In parallel, a **Local-Discriminative Contrastive** module constructs self-supervised views, projects embeddings into a contrastive space, and optimizes an InfoNCE loss to enhance instance-level discrimination and maintain

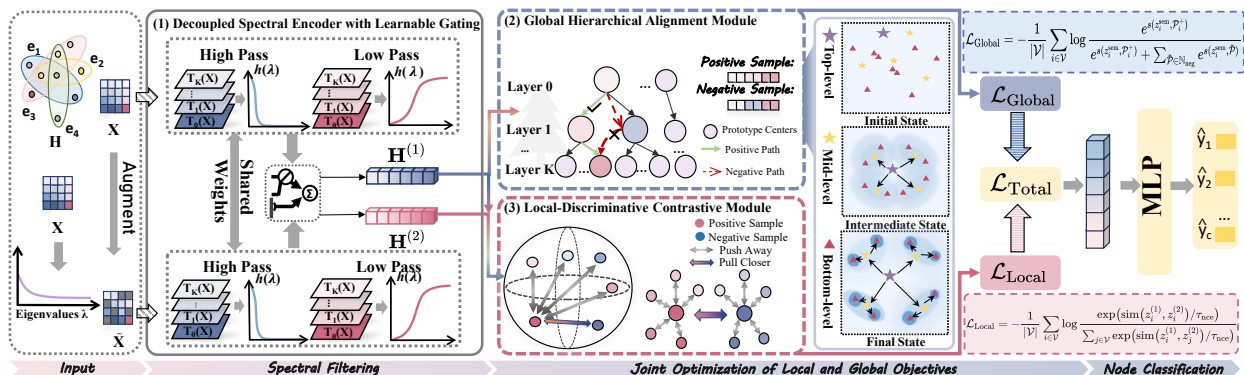

*Figure 2.* Schematic of the proposed HyperDepth framework.

hyperspherical uniformity. Finally, the two modules are coupled through a joint optimization that balances the global semantic and local discriminative objectives. A rigorous theoretical analysis of key properties is provided in Section 6.

### 5.2. Decoupled Spectral Encoder with Adaptive Gating

**Dual-band spectral filtering.** We first perform dual-band spectral filtering based on the normalized hypergraph Laplacian $\mathbf{L}$, defining two responses

$$g_{\text{low}}(\lambda) = e^{-\beta\lambda}, \qquad g_{\text{high}}(\lambda) = 1 - e^{-\beta\lambda},$$

where $\beta > 0$ is a bandwidth parameter controlling the transition between low and high frequencies: eigenvalues close to 0 are mostly passed by $g_{\text{low}}$ and suppressed by $g_{\text{high}}$, while large eigenvalues exhibit the opposite behavior. To avoid the prohibitive cost of matrix exponentials, we implement $g_{\text{low}}$ and $g_{\text{high}}$ via order-$R$ Chebyshev polynomial approximations (Hammond et al., 2011), which reduces the filtering complexity to $O(R|\mathcal{E}|)$. Applying these filters to the input features yields a low-pass representation $h_{\text{low},i}$ and a high-pass representation $h_{\text{high},i}$ for each node $i$. The low-pass branch captures components associated with small eigenvalues and is responsible for preserving *global-semantic structure*, while the high-pass branch captures large-eigenvalue components that emphasize *local-discriminative structure* by highlighting variations relative to neighboring nodes.

**Adaptive spectral gating.** A fixed linear combination of the two branches cannot accommodate the heterogeneous roles of nodes in different regions of the hypergraph. We therefore introduce a learnable *adaptive gating* mechanism to fuse them in a data-dependent manner. For node $i$, the gating coefficient $\alpha_i \in [0,1]$ is computed by a two-layer MLP:

$$\alpha_i = \sigma(\mathbf{W}_2 \operatorname{ReLU}(\mathbf{W}_1[h_{\text{high},i} \| h_{\text{low},i}])), \quad (3)$$

where $[\cdot\|\cdot]$ denotes concatenation, $\mathbf{W}_1 \in \mathbb{R}^{d\times 2d}$ and $\mathbf{W}_2 \in \mathbb{R}^{1\times d}$ are learnable weight matrices, and $\sigma(\cdot)$ is the sigmoid function. The final spectrally enriched embedding for node $i$ is then given by

$$h_i = \alpha_i\, h_{\text{high},i} + (1 - \alpha_i)\, h_{\text{low},i}. \quad (4)$$

This gating mechanism allows HyperDepth to adaptively emphasize high-frequency (local-discriminative) or low-frequency (global-semantic) components for different nodes, providing a flexible spectral basis for the subsequent global alignment and local contrastive objectives.

### 5.3. Global Hierarchical Alignment

To capture the *global-semantic structure* and counteract *Semantic Flattening* as introduced in Section 4, HyperDepth introduces an energy-based alignment module that pulls semantically related nodes towards a shared hierarchy of prototypes while preserving semantic depth.

**Hierarchical prototypes and alignment energy.** We maintain a set of learnable prototypes $\mathcal{C} = \{\mathcal{C}^{(1)}, \ldots, \mathcal{C}^{(K)}\}$, where $\mathcal{C}^{(k)}$ denotes the prototypes at level $k$ of a tree-structured semantic hierarchy and $|\mathcal{C}^{(k)}| < |\mathcal{C}^{(k+1)}|$. Each node $i$ is associated with a semantic path $\mathcal{P}_i = (c_i^{(1)}, \ldots, c_i^{(K)})$ from root to leaf. We define the alignment energy for a node–path pair as

$$E(h_i, \mathcal{P}_i) = -\left(h_i^\top c_i^{(K)} + \lambda_{\text{coh}} \sum_{k=1}^{K-1} (c_i^{(k)})^\top c_i^{(k+1)}\right), \quad (5)$$

where $h_i$ is the spectrally enriched representation and $\lambda_{\text{coh}}$ controls the strength of hierarchical regularization. The instance–leaf term $h_i^\top c_i^{(K)}$ encourages compact fine-grained clusters, while the path-coherence term $\sum_k (c_i^{(k)})^\top c_i^{(k+1)}$ regularizes transitions along the path so that nodes sharing ancestors remain organized in a nested manner rather than being flattened into a single layer.

**Geometric rectification of prototypes.** The prototypes $\mathcal{C}$ are updated jointly with the encoder by backpropagation through Eq. (5). To avoid degenerate placements caused by the non-convex landscape, we periodically perform a lightweight *geometric rectification* step that approximates an EM-style centroid update. Given current embeddings $\{h_i\}$ and a soft assignment $\pi_{i,m}^{(k)} \propto \exp(h_i^\top c_m^{(k)})$ of node $i$ to prototype $c_m^{(k)} \in \mathcal{C}^{(k)}$, we recompute prototypes as

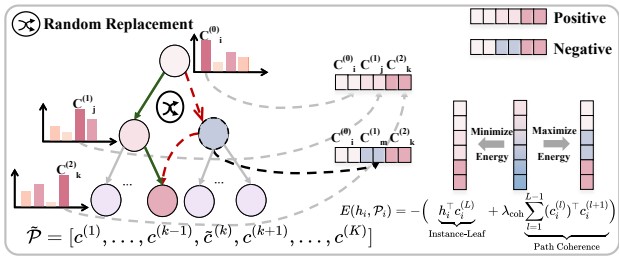

*Figure 3.* Energy-based hierarchical alignment in HyperDepth. Nodes align to positive prototype paths (green), while negative paths (red) are created by random prototype replacement at one level and optimized with an NCE loss.

$$c_m^{(k)} \leftarrow \frac{\sum_i \pi_{i,m}^{(k)} h_i}{\sum_i \pi_{i,m}^{(k)}}, \qquad k = 1, \ldots, K. \tag{6}$$

This update shifts each prototype towards the (weighted) center of the embeddings that support it, effectively recentering prototypes on high-density regions in the embedding space before subsequent gradient-based refinement.

**Energy-based training with path perturbation.** To optimize hierarchical alignment, we employ an energy-based NCE objective on node–path pairs, as illustrated in Figure 3. Instead of operating directly in the shared representation space, we *decouple* semantic alignment from instance discrimination by mapping the representation $h_i^{(1)}$ to a dedicated semantic subspace via a projection head $g_{\text{sem}}(\cdot)$, and define $z_i^{\text{sem}} = g_{\text{sem}}(h_i^{(1)})$. Consistent with Eq. (5), we reuse the energy with $z_i^{\text{sem}}$ in place of $h_i$ and define the alignment score

$$s\big(z_i^{\text{sem}}, \mathcal{P}\big) = -\frac{E\big(z_i^{\text{sem}}, \mathcal{P}\big)}{\tau_{\text{proto}}}, \tag{7}$$

where $\tau_{\text{proto}}$ is a temperature. For each node $i$, we first obtain a positive semantic path $\mathcal{P}_i^+$ by a top-down greedy search that, at each level, selects the prototype maximizing the local contribution to Eq. (5), resulting in an $O(K\bar{B})$ inference cost ($\bar{B}$ is the average branching factor of the prototype tree, see details in Appendix C). To generate hard but structured negatives, we perform *path perturbation*: starting from $\mathcal{P}_i^+$, we randomly choose a level $k$ and replace $c^{(k)}$ with another prototype $\tilde{c}^{(k)} \in \mathcal{C}^{(k)} \setminus \{c^{(k)}\}$, obtaining a perturbed path $\tilde{\mathcal{P}}$ that differs from $\mathcal{P}_i^+$ at exactly one level while preserving all other ancestors and descendants. Collecting such negatives into a set $\mathbb{N}_{\text{neg}}$, the global loss is defined as

$$\mathcal{L}_{\text{Global}} = -\frac{1}{|\mathcal{V}|} \sum_{i \in \mathcal{V}} \log \frac{e^{s(z_i^{\text{sem}}, \mathcal{P}_i^+)}}{e^{s(z_i^{\text{sem}}, \mathcal{P}_i^+)} + \sum_{\tilde{\mathcal{P}} \in \mathbb{N}_{\text{neg}}} e^{s(z_i^{\text{sem}}, \tilde{\mathcal{P}})}}.$$

Minimizing $\mathcal{L}_{\text{Global}}$ pushes each node into the "energy valley" defined by its hierarchical prototype path while repelling it from nearby, minimally perturbed paths, thereby sharpening decision boundaries between prototype hierarchies and recovering semantic depth in the embedding space.

## 5.4. Local-Discriminative Contrastive Learning

To complement global hierarchical alignment, HyperDepth equips the encoder with a contrastive branch that emphasizes high-frequency variations and confines hyperspherical uniformity to a dedicated projection space, thereby instantiating the local-discriminative objective introduced in Section 4.

**High-frequency aware view generation.** The decoupled spectral encoder in Section 5.2 produces node representations in which high-frequency components capture local deviations relative to neighboring nodes, providing a natural signal for instance discrimination. To construct self-supervised views, we generate two perturbed feature matrices $\tilde{\mathbf{X}}^{(1)}$ and $\tilde{\mathbf{X}}^{(2)}$ by applying feature masking and Gaussian noise to the original features $\mathbf{X}$, while keeping the hypergraph structure fixed. Passing these through the shared encoder $f_\Phi$ yields two sets of node embeddings:

$$\mathbf{H}^{(1)} = f_\Phi(\tilde{\mathbf{X}}^{(1)}), \qquad \mathbf{H}^{(2)} = f_\Phi(\tilde{\mathbf{X}}^{(2)}),$$

where $h_i^{(v)}$ denotes the embedding of node $i$ in view $v \in \{1, 2\}$. These embeddings retain the high-frequency information produced by the spectral encoder and serve as the basis for enforcing node-level uniqueness.

**Projection head and local objective.** To reconcile instance-level uniformity with hierarchical alignment, we separate the spaces on which the two objectives act. The global module operates in the representation space (and its semantic subspace), while local discrimination is enforced only after a non-linear projection. Concretely, we adopt a projection head $g_{\text{loc}}(\cdot)$ that maps $h_i^{(v)}$ into a contrastive space, i.e., $z_i^{(v)} = g_{\text{loc}}(h_i^{(v)})$, and apply an InfoNCE loss on $\{z_i^{(1)}, z_i^{(2)}\}$. Using cosine similarity $\text{sim}(\cdot, \cdot)$ and temperature $\tau_{\text{nce}}$, the local loss is

$$\mathcal{L}_{\text{Local}} = -\frac{1}{|\mathcal{V}|} \sum_{i \in \mathcal{V}} \log \frac{\exp\big(\text{sim}(z_i^{(1)}, z_i^{(2)})/\tau_{\text{nce}}\big)}{\sum_{j \in \mathcal{V}} \exp\big(\text{sim}(z_i^{(1)}, z_j^{(2)})/\tau_{\text{nce}}\big)}.$$

This objective pulls together the two augmented views of the same node while pushing them away from other nodes, thereby realizing the hyperspherical uniformity principle in the projection space. Crucially, because $\mathcal{L}_{\text{Local}}$ acts only on $z_i^{(v)}$ while the global hierarchical alignment operates on $h_i$ and its semantic projection, the uniformity pressure is constrained to the contrastive head rather than directly flattening the representation space. As a result, HyperDepth can simultaneously maintain strong node-level discriminability and preserve the semantic depth induced by the prototype hierarchy.

## 5.5. Model Training

HyperDepth reconciles the Hierarchy–Uniformity Conflict by optimizing local and global objectives on different spaces:

the contrastive loss $\mathcal{L}_{\text{Local}}$ acts on the projected embeddings $z$ to promote high-frequency, instance-level uniformity, while the alignment loss $\mathcal{L}_{\text{Global}}$ acts on the representation/semantic space $h$ to preserve low-frequency hierarchical structure. The overall training objective is

$$\mathcal{L}_{\text{Total}} = \mathcal{L}_{\text{Local}} + \gamma \mathcal{L}_{\text{Global}}, \qquad (8)$$

where $\gamma$ controls the trade-off between instance uniqueness and semantic depth.

## 6. Theoretical Analysis

### 6.1. Subspace Decomposition of Local & Global Losses

Let $\mathbf{P}_{\text{low}}$ and $\mathbf{P}_{\text{high}}$ denote the orthogonal projectors onto the low- and high-frequency eigenspaces, respectively, with $\mathbf{P}_{\text{low}} + \mathbf{P}_{\text{high}} = \mathbf{I}$. For each node embedding $h_i \in \mathbb{R}^d$, we write $h_i = h_i^{\text{low}} + h_i^{\text{high}}, h_i^{\text{low}} = \mathbf{P}_{\text{low}} h_i, h_i^{\text{high}} = \mathbf{P}_{\text{high}} h_i$.

**Assumption 1** (Subspace separation). *The local projection head $g_{\text{loc}}$ depends only on $h_i^{\text{high}}$, i.e. $g_{\text{loc}}(h_i) = g_{\text{loc}}(h_i^{\text{high}})$, and the semantic projection head $g_{\text{sem}}$ and the alignment energy in Eq. (5) depend only on $h_i^{\text{low}}$, i.e. $g_{\text{sem}}(h_i) = g_{\text{sem}}(h_i^{\text{low}})$ and $E(h_i, \mathcal{P}) = E(g_{\text{sem}}(h_i^{\text{low}}), \mathcal{P})$.*

**Theorem 1** (Subspace-wise optimization). *Under Assumption 1, the total loss*

$$\mathcal{L}_{\text{Total}} = \mathcal{L}_{\text{Local}}(g_{\text{loc}}(h^{\text{high}})) + \gamma \mathcal{L}_{\text{Global}}(g_{\text{sem}}(h^{\text{low}}))$$

*satisfies, for every node $i$,*

$$\nabla_{h_i} \mathcal{L}_{\text{Total}} = \mathbf{P}_{\text{high}} \nabla_{h_i} \mathcal{L}_{\text{Local}} + \gamma \mathbf{P}_{\text{low}} \nabla_{h_i} \mathcal{L}_{\text{Global}}.$$

*In particular, at any stationary point $\{h_i, \mathcal{C}\}$ of $\mathcal{L}_{\text{Total}}$, the high-frequency components $h_i^{\text{high}}$ are stationary for $\mathcal{L}_{\text{Local}}$ restricted to the high-frequency subspace, and the low-frequency components $h_i^{\text{low}}$ are stationary for $\mathcal{L}_{\text{Global}}$ restricted to the low-frequency subspace.*

**Definition 2** (Hierarchy–uniformity equilibrium). *An embedding $\{h_i\}$ and prototype set $\mathcal{C}$ are said to be in a hierarchy–uniformity equilibrium if i) $\{h_i^{\text{low}}\}$ minimizes $\mathcal{L}_{\text{Global}}(g_{\text{sem}}(h^{\text{low}}))$ over feasible embeddings; ii) $\{h_i^{\text{high}}\}$ minimizes $\mathcal{L}_{\text{Local}}(g_{\text{loc}}(h^{\text{high}}))$ over feasible embeddings.*

**Assumption 2** (Smooth hierarchical semantics). *There exists an embedding $\{h_i^\star\}$ and a prototype hierarchy such that: i) nodes sharing the same leaf prototype have identical low-frequency components, i.e. $h_i^{\star,\text{low}} = h_j^{\star,\text{low}}$ whenever $i$ and $j$ share a leaf; ii) $\{h_i^{\star,\text{low}}\}$ achieves the minimum of $\mathcal{L}_{\text{Global}}(g_{\text{sem}}(h^{\text{low}}))$ over feasible embeddings.*

**Theorem 2** (Existence of hierarchy–uniformity equilibrium). *Suppose Assumptions 1 and 2 hold, and there exists a set of high-frequency components $\{\tilde{h}_i^{\text{high}}\}$ that (approximately) minimizes $\mathcal{L}_{\text{Local}}(g_{\text{loc}}(h^{\text{high}}))$ in the high-frequency subspace. Then the embedding $h_i = h_i^{\star,\text{low}} + \tilde{h}_i^{\text{high}}$ together with the corresponding optimal prototypes $\mathcal{C}$ constitutes a hierarchy–uniformity equilibrium in the sense of Definition 2.*

### 6.2. Semantic Depth Induced by Hierarchical Energy

We now characterize how the hierarchical energy (Eq. (5)) shapes semantic depth in the low-frequency space. For simplicity, assume that $g_{\text{sem}}$ is the identity map and that all embeddings and prototypes are $\ell_2$-normalized. Let $\mu_\ell$ denote the mean direction of embeddings belonging to leaf $\ell$ of the prototype tree, and let $\text{desc}(u)$ denote the set of descendant leaves of an internal node $u$.

**Proposition 1** (Prototype hierarchy preserves semantic depth). *Fix the tree-structured prototype hierarchy and $\lambda_{\text{coh}} > 0$. Suppose that for each leaf $\ell$, all nodes assigned to $\ell$ share the same embedding direction $h_i = \mu_\ell$. Then, at any global minimizer of the global alignment loss $\mathcal{L}_{\text{Global}}$: i) each leaf prototype $c_\ell^{(K)}$ is aligned with its leaf mean, i.e. $c_\ell^{(K)} \parallel \mu_\ell$ for all leaves $\ell$; ii) each internal prototype $c_u$ is aligned with a weighted average of the mean directions of its descendant leaves, i.e., $c_u \parallel \sum_{\ell \in \text{desc}(u)} w_\ell \mu_\ell, w_\ell \geq 0$.*

Full proofs and further generalizations are provided in **Appendix B**.

## 7. Experiments

We conduct extensive experiments on node classification to evaluate HyperDepth and answer the following research questions: i) **RQ1:** How effective is HyperDepth compared with strong supervised and self-supervised baselines across homophilic and heterophilic hypergraph benchmarks? ii) **RQ2:** How much does each key component of HyperDepth contribute to the overall performance? iii) **RQ3:** What structures do the learned hierarchical prototypes exhibit, and do visualizations of these prototypes and embeddings support the advantages of the full objective $\mathcal{L}_{\text{Total}}$? iv) **RQ4:** How sensitive is HyperDepth to key hyperparameters?

### 7.1. Datasets

We evaluate HyperDepth on a diverse collection of real-world hypergraph datasets grouped into two categories. **Homophilic** datasets include citation networks (Cora-C, Citeseer, Pubmed) (Sen et al., 2008), the co-authorship dataset Cora-A (Rossi & Ahmed, 2015), computer vision and graphics benchmarks NTU2012 (Chen et al., 2003) and ModelNet40 (Wu et al., 2015), and the Zoo, 20News, and Mushroom datasets provided by HyFi (Roh et al., 2024). **Heterophilic** datasets include Actor, Amazon-Ratings, Twitch, and Pokec from Li et al. (2025b), as well as House and Senate (Fowler, 2006). Detailed statistics for all datasets are given in **Appendix D**.

### 7.2. Baselines and Experimental Setup

We compare HyperDepth against a broad set of supervised and self-supervised baselines. The *supervised* methods in-

*Table 1.* Node classification accuracy (%) on homophilic hypergraph benchmarks. Best results are shown in **bold** font and second-best results in blue . '-' indicates runs that did not finish within 24 hours. A.R. denotes the average rank over all datasets.

| | Method | Cora-C | Citeseer | Pubmed | Cora-A | Zoo | 20News | Mushroom | NTU2012 | ModelNet40 | A.R. |
|---|---|---|---|---|---|---|---|---|---|---|---|
| Supervised | MLP | 60.32 ± 1.5 | 62.06 ± 2.3 | 76.27 ± 1.1 | 64.05 ± 1.4 | 75.62 ± 9.5 | 78.19 ± 0.5 | 99.58 ± 0.3 | 65.17 ± 2.3 | 93.75 ± 0.6 | 13.11 |
| | GCN | 77.11 ± 1.8 | 66.07 ± 2.4 | 82.63 ± 0.6 | 73.66 ± 1.3 | 36.79 ± 9.6 | 79.41 ± 0.3 | 92.47 ± 0.9 | 71.17 ± 2.4 | 91.67 ± 0.2 | 12.56 |
| | GAT | 77.75 ± 2.1 | 67.62 ± 2.5 | 81.96 ± 0.7 | 74.52 ± 1.3 | 36.48 ± 10.0 | 79.54 ± 0.3 | 95.09 ± 0.5 | 70.94 ± 2.6 | 91.43 ± 0.3 | 12.11 |
| | HGNN | 77.50 ± 1.8 | 66.16 ± 2.3 | 83.52 ± 0.7 | 74.38 ± 1.2 | 78.58 ± 11.1 | 80.15 ± 0.3 | 98.59 ± 0.5 | 72.03 ± 2.4 | 92.23 ± 0.2 | 8.33 |
| | HyperConv | 76.19 ± 2.1 | 64.12 ± 2.6 | 83.42 ± 0.6 | 73.52 ± 1.0 | 62.53 ± 14.5 | 79.83 ± 0.4 | 97.56 ± 0.6 | 72.62 ± 2.6 | 91.84 ± 0.1 | 10.67 |
| | HNHN | 76.21 ± 1.7 | 67.28 ± 2.2 | 80.97 ± 0.9 | 74.88 ± 1.6 | 78.89 ± 10.2 | 79.51 ± 0.4 | **99.78 ± 0.1** | 71.45 ± 3.2 | 92.96 ± 0.2 | 9.11 |
| | HyperGCN | 64.11 ± 7.4 | 59.92 ± 9.6 | 78.40 ± 9.2 | 60.65 ± 9.2 | 40.86 ± 2.1 | 77.31 ± 6.0 | 48.26 ± 3.9 | 46.05 ± 3.9 | 69.23 ± 2.8 | 16.39 |
| | HyperSAGE | 64.98 ± 5.3 | 52.43 ± 9.4 | 79.49 ± 8.7 | 64.59 ± 4.3 | 40.86 ± 2.1 | – | – | – | – | 17.11 |
| | UniGCN | 77.91 ± 1.9 | 66.40 ± 1.9 | 84.08 ± 0.7 | 77.30 ± 1.4 | 72.10 ± 12.1 | 80.24 ± 0.4 | 98.84 ± 0.5 | 73.27 ± 2.7 | 94.62 ± 0.2 | 8.33 |
| | AllSetTransformer | 76.21 ± 1.7 | 67.83 ± 1.8 | 82.85 ± 0.9 | 76.94 ± 1.3 | 72.72 ± 11.8 | 79.90 ± 0.4 | 99.48 ± 0.3 | 75.09 ± 2.5 | 96.85 ± 0.2 | 6.50 |
| Self-Supervised | Random-Init | 63.62 ± 3.1 | 60.44 ± 2.5 | 67.49 ± 2.2 | 66.27 ± 2.2 | 78.43 ± 11.0 | 77.14 ± 0.6 | 97.40 ± 0.6 | 74.39 ± 2.6 | 96.29 ± 0.3 | 12.22 |
| | Node2vec | 70.99 ± 1.4 | 53.85 ± 1.9 | 78.75 ± 0.9 | 58.50 ± 2.1 | 17.02 ± 4.1 | 63.35 ± 1.7 | 88.16 ± 0.8 | 67.72 ± 2.1 | 84.94 ± 0.4 | 16.78 |
| | DGI | 78.17 ± 1.4 | 68.81 ± 1.8 | 80.83 ± 0.6 | 76.94 ± 1.1 | 36.54 ± 9.7 | 73.51 ± 0.7 | 96.71 ± 0.5 | 72.01 ± 2.5 | 92.18 ± 0.2 | 10.83 |
| | GRACE | 79.11 ± 1.7 | 68.65 ± 1.7 | 80.08 ± 0.7 | 76.59 ± 1.0 | 37.07 ± 9.3 | – | 96.01 ± 0.4 | 70.51 ± 2.4 | 90.68 ± 0.3 | 12.33 |
| | S2-HHGR | 78.08 ± 1.7 | 68.21 ± 1.8 | 82.13 ± 0.6 | 78.15 ± 1.1 | 80.06 ± 11.1 | 79.75 ± 0.3 | 97.15 ± 0.5 | 73.95 ± 2.4 | 93.26 ± 0.2 | 7.33 |
| | TriCL | 81.03 ± 1.3 | 71.97 ± 1.3 | 83.80 ± 0.6 | **81.02 ± 1.0** | 79.47 ± 11.0 | 79.93 ± 0.2 | 98.93 ± 0.3 | 74.63 ± 2.5 | 97.33 ± 0.1 | 3.56 |
| | MMACL | 80.60 ± 1.2 | 72.30 ± 1.2 | 82.60 ± 0.6 | 79.50 ± 1.4 | 74.90 ± 1.7 | – | – | 74.91 ± 1.7 | 96.94 ± 1.5 | 8.06 |
| | HyFi | 81.48 ± 1.5 | 72.57 ± 1.1 | 83.82 ± 0.6 | 79.33 ± 1.3 | 80.02 ± 10.9 | 79.76 ± 0.3 | 99.65 ± 0.2 | 75.10 ± 2.6 | 97.38 ± 0.1 | 3.22 |
| | HyperAim | 81.08 ± 1.4 | **73.13 ± 1.2** | 83.02 ± 0.5 | 78.78 ± 1.4 | 80.31 ± 4.8 | 79.83 ± 0.2 | 99.68 ± 0.4 | 74.76 ± 1.5 | 97.37 ± 0.1 | 3.89 |
| | **HyperDepth** | **82.69 ± 1.2** | 72.80 ± 1.4 | 83.48 ± 0.4 | 80.71 ± 1.2 | **89.01 ± 2.6** | **80.87 ± 0.4** | 99.71 ± 0.4 | **79.34 ± 0.8** | **97.61 ± 0.2** | **1.89** |

clude GCN (Kipf & Welling, 2017), GAT (Veličković et al., 2018), HGNN (Feng et al., 2019), HyperConv (Bai et al., 2021), HNHN (Dong et al., 2020), HyperGCN (Yadati et al., 2019), HyperSAGE (Arya et al., 2020), UniGCN (Huang & Yang, 2021), and AllSetTransformer (Chien et al., 2022). The *self-supervised* baselines include Node2vec (Grover & Leskovec, 2016), DGI (Veličković et al., 2019), GRACE (Zhu et al., 2020), S2-HHGR (Zhang et al., 2021), TriCL (Lee & Shin, 2023), MMACL (Lee & Chae, 2024), HyFi (Roh et al., 2024), HyperAim (Li et al., 2026). For graph-based methods: GCN (Kipf & Welling, 2017), DGI (Veličković et al., 2019), GRACE (Zhu et al., 2020), we follow prior work and apply them to clique-expanded graphs.

For evaluation, we adopt the linear probing protocol used in HyFi (Roh et al., 2024). We first train each self-supervised method (including HyperDepth) to obtain node representations, freeze the encoder, and then train a linear classifier on randomly generated 10/10/80 train/validation/test splits. Unless otherwise specified, all methods share the same splits. Further implementation details and dataset-specific hyperparameters for reproducibility are reported in **Appendix D**.

### 7.3. Overall Performance Comparison (RQ1)

Tables 1 and 2 report the node classification results on homophilic and heterophilic datasets.

On the homophilic datasets (Table 1), HyperDepth consistently ranks among the top methods and achieves the best average rank across all methods, including fully supervised hypergraph neural networks such as AllSetTransformer and UniGCN as well as strong self-supervised baselines such as HyFi and HyperAim. In particular, HyperDepth attains the best or second-best performance on most datasets while

using only self-supervised training, showing that explicitly resolving the hierarchy–uniformity conflict can close and often surpass the gap to label-supervised models.

In particular, for the heterophilic benchmarks as shown in Table 2, where structural assumptions are more challenging, HyperDepth further strengthens its advantage within the hypergraph contrastive learning family. Compared with recent self-supervised methods tailored to hypergraphs (HyFi, MMACL, HyperAim), HyperDepth obtains the best performance on five out of six datasets and remains competitive on the remaining one.

*Table 2.* Performance comparison on heterophilic datasets.

| Method | Actor | Amazon | Twitch | Pokec | House | Senate |
|---|---|---|---|---|---|---|
| Node2vec | 59.0 ± 0.3 | 25.1 ± 0.3 | 50.0 ± 0.6 | 50.5 ± 0.5 | 50.9 ± 1.0 | 48.2 ± 2.7 |
| DGI | 61.5 ± 0.4 | 25.3 ± 0.3 | 51.2 ± 0.4 | 52.9 ± 0.5 | 54.9 ± 1.2 | 53.6 ± 2.6 |
| GRACE | 62.3 ± 0.5 | 25.7 ± 0.2 | 50.1 ± 0.4 | 56.0 ± 0.7 | 55.9 ± 1.5 | 49.3 ± 1.3 |
| S2-HHGR | 62.3 ± 0.6 | 19.0 ± 0.2 | 50.0 ± 0.3 | 49.2 ± 0.3 | 50.1 ± 1.2 | 49.6 ± 1.5 |
| TriCL | 65.2 ± 0.5 | 26.4 ± 0.4 | 50.6 ± 0.5 | 55.4 ± 0.9 | 74.4 ± 1.1 | 58.2 ± 3.6 |
| HyFi | 62.3 ± 0.2 | 19.1 ± 0.1 | 50.0 ± 0.2 | 49.2 ± 0.2 | 73.7 ± 1.1 | 66.5 ± 2.6 |
| MMACL | – | – | – | – | 72.7 ± 1.1 | 63.8 ± 2.6 |
| HyperAim | 81.0 ± 0.5 | **29.2 ± 0.3** | 51.5 ± 0.4 | 57.0 ± 0.7 | 74.7 ± 1.9 | 75.3 ± 0.6 |
| **HyperDepth** | **83.1 ± 0.2** | 28.7 ± 0.3 | **52.0 ± 0.6** | **57.2 ± 0.5** | **76.1 ± 0.9** | **76.2 ± 1.8** |

### 7.4. Ablation Studies (RQ2)

We investigate four variants of HyperDepth: removing the local contrastive loss (**w/o** $\mathcal{L}_{\text{Local}}$), removing the global hierarchical loss (**w/o** $\mathcal{L}_{\text{Global}}$), disabling the low-pass branch (**w/o low-pass**), and disabling the high-pass branch (**w/o high-pass**).

As shown in Figure 5, the full model reaches 82.69% accuracy on Cora-C, while **w/o** $\mathcal{L}_{\text{Local}}$ drops to 78.93% and **w/o** $\mathcal{L}_{\text{Global}}$ to 80.50%, indicating that both the local contrastive objective and the global hierarchical alignment are important, in line with our **Theorem 1** showing that the two

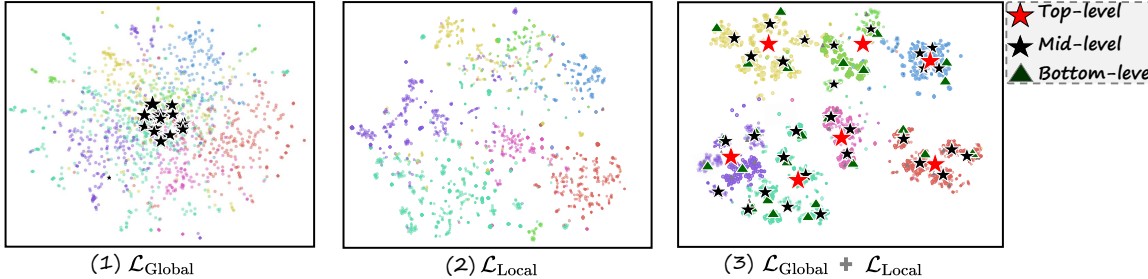

(1) $\mathcal{L}_{\text{Global}}$    (2) $\mathcal{L}_{\text{Local}}$    (3) $\mathcal{L}_{\text{Global}} + \mathcal{L}_{\text{Local}}$

*Figure 4.* Visualization of Hierarchical Prototypes. The distribution of Red (Top-level), Black (Mid-level), and **Green** (Bottom-level) prototypes demonstrates how HyperDepth captures multi-scale semantic structures.

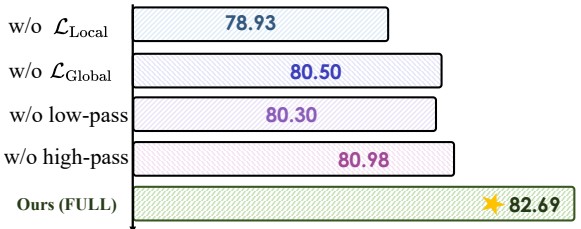

*Figure 5.* Ablation study on HyperDepth.

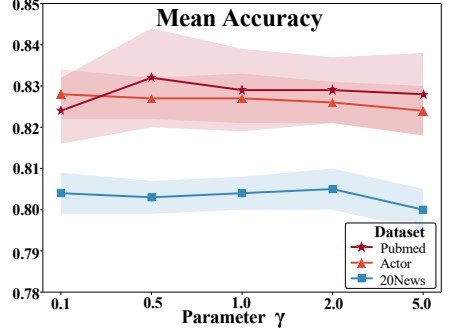

*Figure 6.* Sensitivity of the balance parameter $\gamma$ on three datasets.

objectives jointly admit equilibrium embeddings rather than competing in the same space. On the spectral side, removing the low-pass channel (80.30%) is more detrimental than removing high frequencies (80.98%), which is consistent with **Theorem 2** that the local and global terms operate on complementary spectral components with low-frequency structure dominating on homophilic graphs. Similar trends are observed on other benchmarks, and additional ablation results are reported in **Appendix D**.

### 7.5. Visualization of Hierarchical Prototypes (RQ3)

We visualize the learned embeddings and hierarchical prototypes on a representative dataset in Figure 4. With only the global loss $\mathcal{L}_{\text{Global}}$ (**panel (1)**), prototypes tend to collapse toward the center and provide little guidance for separating node clusters, illustrating that hierarchy alone is not sufficient to shape a discriminative geometry. With only the local loss $\mathcal{L}_{\text{Local}}$ (**panel (2)**), clusters become well separated but lack clear multi-scale centroids, and no coherent prototype hierarchy emerges, reflecting the *uniformity-dominated be-*

*havior* discussed in Section 4. In contrast, the full objective $\mathcal{L}_{\text{Total}}$ (**panel (3)**) yields a structured latent space where top-, mid-, and bottom-level prototypes ( red stars, black stars, and green triangles) are positioned near the corresponding groups of nodes and form a consistent tree-like layout. This empirical pattern is consistent with Theorem 1: combining the local and global objectives leads to an equilibrium embedding that preserves both semantic depth (captured by the prototype tree) and instance-level discriminability (visible in the separated clusters).

### 7.6. Hyperparameter Sensitivity Analysis (RQ4)

We study the effect of the balance parameter $\gamma$ on three representative datasets by varying $\gamma \in \{0.1, 0.5, 1.0, 2.0, 5.0\}$. As shown in Figure 6, HyperDepth exhibits stable behaviour across three datasets (i.e., Pubmed, Actor, 20News), that is, the mean accuracy changes only slightly over this range and typically peaks around $\gamma \approx 1.0$, where the local and global objectives receive comparable weights. This indicates that HyperDepth does not require meticulous tuning of $\gamma$ and that a moderate trade-off between local discrimination and hierarchical alignment is sufficient in practice. Additional sensitivity results for other hyperparameters (e.g., $R$, $K$, and temperature $\tau$) over all 15 datasets are provided in **Appendix D**.

## 8. Conclusion

We introduce the hierarchy–uniformity conflict in hypergraph contrastive learning and propose HyperDepth to explicitly move representations towards a hierarchy–uniformity equilibrium. HyperDepth combines a decoupled spectral encoder with adaptive gating and an energy-based hierarchical alignment module that attaches a prototype tree to the representation space. Our theoretical analysis shows that the local contrastive and global hierarchical objectives act on complementary spectral components and admit equilibrium embeddings that preserve both semantic depth and instance-level discrimination. Experiments on homophilic and heterophilic hypergraph benchmarks support the effectiveness of the proposed framework.

## Impact Statement

This paper aims to advance research in Hypergraph Contrastive Learning, with broader relevance to Hypergraph Machine Learning or Hypergraph Computation. While the primary contributions are methodological and intended for general-purpose representation learning, we do not anticipate immediate societal impacts that warrant specific discussion beyond the standard considerations applicable to machine learning research.

## Acknowledgments

This work was supported in part by the "Pioneer" and "Leading Goose" R&D Program of Zhejiang (No. 2024C03262), and the National Natural Science Foundation of China (No. 62536006, No. 62320106007, No. 62521007, No. 62576371).

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

# A. Notation Summary

We summarize the key notations used throughout the paper in the following Table 3.

*Table 3.* Key notations used in the main text.

| Symbol | Description |
|---|---|
| $\mathcal{H} = (\mathcal{V}, \mathcal{E})$ | Hypergraph with node set $\mathcal{V}$ and hyperedge set $\mathcal{E}$ |
| $N = |\mathcal{V}|, E = |\mathcal{E}|$ | Number of nodes and hyperedges |
| $\mathbf{H} \in \{0,1\}^{N \times E}$ | Incidence matrix, $H_{i,e} = 1$ if node $i$ belongs to hyperedge $e$ |
| $\mathbf{D}_v, \mathbf{D}_e$ | Diagonal matrices of node degrees and hyperedge degrees |
| $\mathbf{L}$ | Normalized hypergraph Laplacian |
| $\mathbf{U}, \mathbf{\Lambda}$ | Eigenvectors and eigenvalues of $\mathbf{L}$, $\mathbf{L} = \mathbf{U}\mathbf{\Lambda}\mathbf{U}^\top$ |
| $\mathbf{P}_{\text{low}}, \mathbf{P}_{\text{high}}$ | Projectors onto low- and high-frequency eigenspaces |
| $k$ | Number of nearest neighbors for initial hypergraph construction |
| $g_{\text{low}}(\lambda), g_{\text{high}}(\lambda)$ | Low- and high-pass spectral response functions |
| $\beta$ | Bandwidth parameter in the spectral filters |
| $R$ | Order of Chebyshev polynomial approximation |
| $\mathbf{X} \in \mathbb{R}^{N \times d_0}$ | Input node feature matrix |
| $f_\Phi$ | HyperDepth encoder with parameters $\Phi$ |
| $\mathbf{H}^{(v)}$ | Encoder outputs for view $v$ |
| $h_i$ | Embedding of node $i$ in representation space |
| $h_i^{\text{low}}, h_i^{\text{high}}$ | Low- and high-frequency components of $h_i$ |
| $g_{\text{loc}}, g_{\text{sem}}$ | Projection heads for local contrastive and semantic branches |
| $z_i^{(v)}$ | Projected embedding of node $i$ in view $v$ (local branch) |
| $z_i^{\text{sem}}$ | Semantic embedding of node $i$ (global branch) |
| $\alpha_i$ | Learnable gating coefficient for node $i$ |
| $\mathbf{W}_1, \mathbf{W}_2$ | Weight matrices of the gating MLP |
| $\mathcal{C}^{(k)}$ | Set of prototypes at hierarchy level $k$ |
| $c^{(k)} \in \mathcal{C}^{(k)}$ | Prototype vector at level $k$ |
| $K$ | Number of hierarchy levels in the prototype tree |
| $\mathcal{P}_i = (c_i^{(1)}, \ldots, c_i^{(K)})$ | Positive prototype path for node $i$ |
| $\tilde{\mathcal{P}}$ | Negative prototype path obtained by path perturbation |
| $\mathbb{N}_{\text{neg}}$ | Set of negative prototype paths for node $i$ |
| $E(h_i, \mathcal{P})$ | Energy of node $i$ with respect to path $\mathcal{P}$ |
| $\lambda_{\text{coh}}$ | Weight of the path-coherence term in the energy |
| $\mathcal{L}_{\text{Local}}$ | Local contrastive loss (InfoNCE) |
| $\mathcal{L}_{\text{Global}}$ | Global energy-based hierarchical alignment loss |
| $\mathcal{L}_{\text{Total}}$ | Overall training objective |
| $\tau_{\text{nce}}, \tau_{\text{proto}}$ | Temperatures for InfoNCE and prototype energy |
| $\gamma$ | Trade-off coefficient between local and global objectives |

# B. Additional Details for Theoretical Analysis

In this part, we provide detailed proofs and additional discussion for the theoretical analysis in Section 6.

## B.1. Spectral Decomposition and Basic Setup

Let $\mathbf{L} = \mathbf{U}\mathbf{\Lambda}\mathbf{U}^\top$ be the eigendecomposition of the normalized hypergraph Laplacian, with $\mathbf{U} = [u_1, \ldots, u_N] \in \mathbb{R}^{N \times N}$ orthogonal and $\mathbf{\Lambda} = \text{diag}(\lambda_1, \ldots, \lambda_N), 0 = \lambda_1 \leq \cdots \leq \lambda_N$.

We partition the eigenvectors into low- and high-frequency parts,

$$\mathbf{U} = [\mathbf{U}_{\text{low}} \ \mathbf{U}_{\text{high}}], \tag{9}$$

where columns in $\mathbf{U}_{\text{low}}$ correspond to the eigenvalues deemed "low-frequency", and columns in $\mathbf{U}_{\text{high}}$ correspond to the remaining "high-frequency" eigenvalues. The associated orthogonal projectors are

$$\mathbf{P}_{\text{low}} = \mathbf{U}_{\text{low}}\mathbf{U}_{\text{low}}^\top, \qquad \mathbf{P}_{\text{high}} = \mathbf{U}_{\text{high}}\mathbf{U}_{\text{high}}^\top. \tag{10}$$

By construction, these satisfy

$$\mathbf{P}_{\text{low}}^2 = \mathbf{P}_{\text{low}}, \quad \mathbf{P}_{\text{high}}^2 = \mathbf{P}_{\text{high}}, \quad \mathbf{P}_{\text{low}}\mathbf{P}_{\text{high}} = \mathbf{0}, \quad \mathbf{P}_{\text{low}} + \mathbf{P}_{\text{high}} = \mathbf{I},$$

and both projectors are symmetric.

For each node embedding $h_i \in \mathbb{R}^d$ we use the decomposition

$$h_i = h_i^{\text{low}} + h_i^{\text{high}}, \quad h_i^{\text{low}} = \mathbf{P}_{\text{low}}h_i, \quad h_i^{\text{high}} = \mathbf{P}_{\text{high}}h_i. \tag{11}$$

Note that $h_i^{\text{low}}$ and $h_i^{\text{high}}$ lie in orthogonal subspaces, and $h_i$ is uniquely determined by their sum.

Throughout the proofs, we treat the encoder parameters, projection heads, and prototypes as fixed when differentiating with respect to $\{h_i\}$.

### B.2. Proof of Theorem 1

*Proof of Theorem 1.* Recall that by Assumption 1, the local projection head and local loss depend only on the high-frequency components:

$$\mathcal{L}_{\text{Local}} = \mathcal{L}_{\text{Local}}\big(g_{\text{loc}}(h^{\text{high}})\big), \quad h_i^{\text{high}} = \mathbf{P}_{\text{high}}h_i.$$

We make the dependence explicit by viewing $\mathcal{L}_{\text{Local}}$ as a function of the collection $h^{\text{high}} = \{h_i^{\text{high}}\}_i$.

For a given node $i$, the dependence of $\mathcal{L}_{\text{Local}}$ on $h_i$ is entirely through $h_i^{\text{high}} = \mathbf{P}_{\text{high}}h_i$. Using the chain rule in vector form,

$$\nabla_{h_i}\mathcal{L}_{\text{Local}} = \left(\frac{\partial h_i^{\text{high}}}{\partial h_i}\right)^{\top} \nabla_{h_i^{\text{high}}}\mathcal{L}_{\text{Local}}. \tag{12}$$

Since $h_i^{\text{high}} = \mathbf{P}_{\text{high}}h_i$ and $\mathbf{P}_{\text{high}}$ is a constant matrix, the Jacobian is

$$\frac{\partial h_i^{\text{high}}}{\partial h_i} = \mathbf{P}_{\text{high}}.$$

Because $\mathbf{P}_{\text{high}}$ is symmetric, we have $\mathbf{P}_{\text{high}}^{\top} = \mathbf{P}_{\text{high}}$, and (12) becomes

$$\nabla_{h_i}\mathcal{L}_{\text{Local}} = \mathbf{P}_{\text{high}} \nabla_{h_i^{\text{high}}}\mathcal{L}_{\text{Local}}. \tag{13}$$

An analogous argument applies to the global loss. Under Assumption 1, the semantic projection head and the alignment energy depend only on the low-frequency components:

$$\mathcal{L}_{\text{Global}} = \mathcal{L}_{\text{Global}}\big(g_{\text{sem}}(h^{\text{low}})\big), \quad h_i^{\text{low}} = \mathbf{P}_{\text{low}}h_i.$$

Again viewing $\mathcal{L}_{\text{Global}}$ as a function of $h^{\text{low}} = \{h_i^{\text{low}}\}_i$, the chain rule gives

$$\nabla_{h_i}\mathcal{L}_{\text{Global}} = \left(\frac{\partial h_i^{\text{low}}}{\partial h_i}\right)^{\top} \nabla_{h_i^{\text{low}}}\mathcal{L}_{\text{Global}} = \mathbf{P}_{\text{low}} \nabla_{h_i^{\text{low}}}\mathcal{L}_{\text{Global}}, \tag{14}$$

since $h_i^{\text{low}} = \mathbf{P}_{\text{low}}h_i$ and $\mathbf{P}_{\text{low}}$ is symmetric.

The total loss is the weighted sum

$$\mathcal{L}_{\text{Total}} = \mathcal{L}_{\text{Local}} + \gamma\mathcal{L}_{\text{Global}},$$

so differentiation with respect to $h_i$ yields

$$\nabla_{h_i}\mathcal{L}_{\text{Total}} = \nabla_{h_i}\mathcal{L}_{\text{Local}} + \gamma\nabla_{h_i}\mathcal{L}_{\text{Global}}.$$

Substituting (13) and (14) gives

$$\nabla_{h_i}\mathcal{L}_{\text{Total}} = \mathbf{P}_{\text{high}} \nabla_{h_i^{\text{high}}}\mathcal{L}_{\text{Local}} + \gamma\mathbf{P}_{\text{low}} \nabla_{h_i^{\text{low}}}\mathcal{L}_{\text{Global}}. \tag{15}$$

To simplify notation, we can interpret $\nabla_{h_i^{\text{high}}}\mathcal{L}_{\text{Local}}$ as $\nabla_{h_i}\mathcal{L}_{\text{Local}}$ computed with respect to the high-frequency component, and similarly for the low-frequency gradient. This yields the expression stated in the theorem:

$$\nabla_{h_i}\mathcal{L}_{\text{Total}} = \mathbf{P}_{\text{high}}\nabla_{h_i}\mathcal{L}_{\text{Local}} + \gamma\,\mathbf{P}_{\text{low}}\nabla_{h_i}\mathcal{L}_{\text{Global}}.$$

We now consider stationary points. If $(\{h_i\}, \mathcal{C})$ is stationary for $\mathcal{L}_{\text{Total}}$, then

$$\nabla_{h_i}\mathcal{L}_{\text{Total}} = 0 \quad \text{for all } i. \tag{16}$$

Left-multiplying (15) by $\mathbf{P}_{\text{high}}$ and using $\mathbf{P}_{\text{high}}\mathbf{P}_{\text{low}} = \mathbf{0}$ yields

$$\mathbf{P}_{\text{high}}\nabla_{h_i}\mathcal{L}_{\text{Total}} = \mathbf{P}_{\text{high}}\mathbf{P}_{\text{high}}\nabla_{h_i}\mathcal{L}_{\text{Local}} + \gamma\,\mathbf{P}_{\text{high}}\mathbf{P}_{\text{low}}\nabla_{h_i}\mathcal{L}_{\text{Global}} = \mathbf{P}_{\text{high}}\nabla_{h_i}\mathcal{L}_{\text{Local}}.$$

Combining this with (16), we obtain

$$\mathbf{P}_{\text{high}}\nabla_{h_i}\mathcal{L}_{\text{Local}} = 0 \quad \text{for all } i,$$

which means that the high-frequency components $\{h_i^{\text{high}}\}$ are stationary for $\mathcal{L}_{\text{Local}}$ when constrained to the high-frequency subspace.

Similarly, left-multiplying (15) by $\mathbf{P}_{\text{low}}$ and using $\mathbf{P}_{\text{low}}\mathbf{P}_{\text{high}} = \mathbf{0}$ gives

$$\mathbf{P}_{\text{low}}\nabla_{h_i}\mathcal{L}_{\text{Total}} = \gamma\,\mathbf{P}_{\text{low}}\nabla_{h_i}\mathcal{L}_{\text{Global}}.$$

At a stationary point this implies

$$\mathbf{P}_{\text{low}}\nabla_{h_i}\mathcal{L}_{\text{Global}} = 0 \quad \text{for all } i,$$

showing that the low-frequency components $\{h_i^{\text{low}}\}$ are stationary for $\mathcal{L}_{\text{Global}}$ in the low-frequency subspace. This completes the proof. □

**Remark 1.** Theorem 1 shows that, under the frequency-separation assumption, the gradients of the local and global objectives are forced into orthogonal spectral components: the local contrastive loss can only modify $h_i^{\text{high}}$, and the global hierarchical loss can only modify $h_i^{\text{low}}$. This formalizes the intuition that HyperDepth reduces direct gradient interference between uniformity and hierarchy, aligning with the design goal of mitigating the Hierarchy–Uniformity Conflict at the representation level.

### B.3. Proof of Theorem 2

*Proof of Theorem 2.* Assumption 2 guarantees the existence of an embedding $\{h_i^\star\}$ such that: (i) nodes sharing the same leaf prototype have identical low-frequency components $h_i^{\star,\text{low}}$, and (ii) the collection $\{h_i^{\star,\text{low}}\}$ minimizes $\mathcal{L}_{\text{Global}}(g_{\text{sem}}(h^{\text{low}}))$ over feasible embeddings.

We first show that the global loss is insensitive to the choice of high-frequency components when the low-frequency parts are fixed to $h_i^{\star,\text{low}}$. Under Assumption 1, the semantic projection head $g_{\text{sem}}$ and the energy term $E(\cdot, \cdot)$ depend only on $h_i^{\text{low}}$, hence for any collection of high-frequency components $\{h_i^{\text{high}}\}$, the embedding

$$h_i = h_i^{\text{low}} + h_i^{\text{high}}$$

satisfies

$$g_{\text{sem}}(h_i) = g_{\text{sem}}(h_i^{\text{low}}), \quad E(h_i, \mathcal{P}) = E(h_i^{\text{low}}, \mathcal{P}),$$

and therefore

$$\mathcal{L}_{\text{Global}}(g_{\text{sem}}(h^{\text{low}}))$$

depends only on $\{h_i^{\text{low}}\}$ and not on $\{h_i^{\text{high}}\}$.

Now consider any embedding of the form

$$h_i = h_i^{\star,\text{low}} + \tilde{h}_i^{\text{high}}, \tag{17}$$

where $\{\tilde{h}_i^{\text{high}}\}$ is an arbitrary set of high-frequency components. For this embedding we have

$$h_i^{\text{low}} = \mathbf{P}_{\text{low}}h_i = \mathbf{P}_{\text{low}}(h_i^{\star,\text{low}} + \tilde{h}_i^{\text{high}}) = \mathbf{P}_{\text{low}}h_i^{\star,\text{low}} + \mathbf{P}_{\text{low}}\tilde{h}_i^{\text{high}} = h_i^{\star,\text{low}},$$

because $h_i^{\star,\text{low}}$ is already in the low-frequency subspace and $\tilde{h}_i^{\text{high}}$ is in the high-frequency subspace. Thus, for the embedding (17), the low-frequency parts are fixed at their globally optimal values, and consequently

$$\mathcal{L}_{\text{Global}}(g_{\text{sem}}(h^{\text{low}})) = \mathcal{L}_{\text{Global}}(g_{\text{sem}}(h^{\star,\text{low}})),$$

so the global loss is minimized.

Next, we consider the local contrastive loss. By Assumption 1, the local projection head $g_{\text{loc}}$ and $\mathcal{L}_{\text{Local}}$ depend only on $h^{\text{high}}$. Hence, for any embedding whose high-frequency components equal a given set $\{\tilde{h}_i^{\text{high}}\}$, we have

$$\mathcal{L}_{\text{Local}}\big(g_{\text{loc}}(h^{\text{high}})\big) = \mathcal{L}_{\text{Local}}\big(g_{\text{loc}}(\tilde{h}^{\text{high}})\big),$$

where $\tilde{h}^{\text{high}} = \{\tilde{h}_i^{\text{high}}\}_i$. By assumption, $\{\tilde{h}_i^{\text{high}}\}$ minimizes (or approximately minimizes) $\mathcal{L}_{\text{Local}}(g_{\text{loc}}(h^{\text{high}}))$ in the high-frequency subspace. Therefore, the embedding (17) achieves the optimal value of the local loss in the high-frequency space and the optimal value of the global loss in the low-frequency space.

By Definition 2, an embedding is in a hierarchy–uniformity equilibrium if the low-frequency components minimize the global loss and the high-frequency components minimize the local loss. We have just shown that the embedding $h_i = h_i^{\star,\text{low}} + \tilde{h}_i^{\text{high}}$ satisfies both properties. This establishes the existence of a hierarchy–uniformity equilibrium under the stated assumptions. $\square$

**Remark 2.** The proof is constructive: one can first solve (conceptually) the global alignment problem in the low-frequency space to obtain $\{h_i^{\star,\text{low}}\}$, and independently solve the local contrastive problem in the high-frequency space to obtain $\{\tilde{h}_i^{\text{high}}\}$. Superimposing these solutions yields an embedding that is optimal with respect to both objectives in their respective subspaces. This formalizes the idea that HyperDepth aims for a representation where contrastive uniformity and hierarchical semantics are not in unavoidable competition.

## B.4. Proof of Proposition 1

For convenience, we recall that the alignment energy for a node $i$ and its prototype path $\mathcal{P}_i$ is

$$E(h_i, \mathcal{P}_i) = -\left(h_i^\top c_i^{(K)} + \lambda_{\text{coh}} \sum_{k=1}^{K-1} (c_i^{(k)})^\top c_i^{(k+1)}\right),$$

with $\lambda_{\text{coh}} > 0$ and all vectors $\ell_2$-normalized. The global loss $\mathcal{L}_{\text{Global}}$ is defined via an NCE objective over positive and negative paths; here we focus on the structure induced at energy minima, i.e., on the behavior of the positive paths.

*Proof of Proposition 1.* We prove the two claims separately: leaf prototypes and internal prototypes.

**Leaf-level prototypes.** Fix a leaf $\ell$ in the prototype tree and consider all nodes whose positive path ends at $\ell$. By assumption, all such nodes share the same embedding direction $h_i = \mu_\ell$ with $\|\mu_\ell\|_2 = 1$. Let $\mathcal{V}_\ell$ denote the set of these nodes.

The contribution of these nodes to the total energy through the instance–leaf term is

$$-\sum_{i \in \mathcal{V}_\ell} h_i^\top c_\ell^{(K)} = -\sum_{i \in \mathcal{V}_\ell} \mu_\ell^\top c_\ell^{(K)} = -|\mathcal{V}_\ell|\, \mu_\ell^\top c_\ell^{(K)}. \tag{18}$$

Because both $\mu_\ell$ and $c_\ell^{(K)}$ are unit-norm, the inner product satisfies $\mu_\ell^\top c_\ell^{(K)} \leq 1$ by Cauchy–Schwarz, with equality if and only if $c_\ell^{(K)}$ is a scalar multiple of $\mu_\ell$, i.e. $c_\ell^{(K)} \parallel \mu_\ell$. Minimizing the energy contribution (18) is equivalent to maximizing $\mu_\ell^\top c_\ell^{(K)}$. Hence, at any global minimizer (of the total energy), we must have

$$c_\ell^{(K)} \parallel \mu_\ell \quad \text{for all leaves } \ell.$$

This proves the first item.

**Internal prototypes.** Now consider an internal node $u$ in the prototype tree and its neighbors in the tree, $\mathcal{N}(u)$ (including its parent and all its children). We keep all other prototypes fixed and examine the dependence of the total energy on $c_u$.

Recall that the path-coherence term for a node $i$ involves inner products between consecutive prototypes along its path. Aggregating over all nodes and paths, each edge $(v, w)$ in the prototype tree contributes a term of the form $-\lambda_{\mathrm{coh}} \, c_v^\top c_w$, whenever the edge $(v, w)$ appears as a consecutive pair along some positive path. For a given internal node $u$, the part of the total energy that depends on $c_u$ can therefore be written as

$$E_u(c_u) = -\lambda_{\mathrm{coh}} \sum_{v \in \mathcal{N}(u)} \alpha_{uv} \, c_v^\top c_u, \tag{19}$$

where $\alpha_{uv} \geq 1$ accounts for the number of times the edge $(u, v)$ is traversed by positive paths (or more generally, for the effective weight of that edge in the energy, which may depend on the NCE formulation). All terms in the global energy that do not involve $c_u$ are constant with respect to $c_u$ and can be ignored for the purpose of minimization over $c_u$.

Since $\lambda_{\mathrm{coh}} > 0$, minimizing (19) is equivalent to maximizing

$$\sum_{v \in \mathcal{N}(u)} \alpha_{uv} \, c_v^\top c_u, \quad \text{subject to } \|c_u\|_2 = 1. \tag{20}$$

Let $m_u = \sum_{v \in \mathcal{N}(u)} \alpha_{uv} \, c_v$. Then the objective in (20) is $m_u^\top c_u$. By Cauchy–Schwarz, $m_u^\top c_u \leq \|m_u\|_2 \|c_u\|_2 = \|m_u\|_2$, with equality if and only if $c_u$ and $m_u$ are aligned. Thus, for fixed neighbors $\{c_v : v \in \mathcal{N}(u)\}$, any global minimizer of the total energy with respect to $c_u$ must satisfy $c_u \parallel m_u = \sum_{v \in \mathcal{N}(u)} \alpha_{uv} \, c_v$. Equivalently, there exist non-negative weights $\tilde{w}_v \geq 0$ such that $c_u \parallel \sum_{v \in \mathcal{N}(u)} \tilde{w}_v \, c_v$.

Combining this with the leaf-level result, we can propagate the structure up the tree: all internal prototypes are aligned with non-negative linear combinations of their neighbors, and the neighbors of leaves include leaf prototypes, which are themselves aligned with leaf means $\mu_\ell$. By recursively substituting this structure from the leaves upwards, we obtain that every internal node $u$ is aligned with a non-negative linear combination of the leaf means of its descendants,

$$c_u \parallel \sum_{\ell \in \mathrm{desc}(u)} w_\ell \mu_\ell, \quad w_\ell \geq 0.$$

Here $\mathrm{desc}(u)$ denotes the set of descendant leaves of $u$, and the weights $\{w_\ell\}$ absorb both the $\alpha_{uv}$ and the recursively propagated coefficients. This establishes the second item of the proposition. $\square$

**Remark 3.** The proposition formalizes the idea that the energy-based prototype hierarchy captures coarse-to-fine semantic structure: leaf prototypes represent fine-grained clusters, while internal prototypes represent increasingly coarse aggregations of their descendant clusters. In particular, the tree does not degenerate into a flat partition—the directions of internal prototypes are constrained to lie in the cone spanned by the descendant leaf means, which encodes how semantic content is gradually merged as we move up the hierarchy.

**Take-home Messages of the Theoretical Analysis.** For convenience, we summarize the main conceptual implications of the results in this appendix.

- **Spectral decoupling reduces gradient interference.** The subspace-wise optimization result (**Theorem 1**) shows that, under frequency separation, gradients from the local contrastive objective and from the global hierarchical objective are confined to orthogonal spectral components. This provides a principled explanation of how HyperDepth mitigates the Hierarchy–Uniformity Conflict at the gradient level.

- **A hierarchy–uniformity equilibrium is feasible.** The existence theorem (**Theorem 2**) demonstrates that, under a mild smoothness assumption on hierarchical semantics, there exist embeddings in which low-frequency components are optimal for hierarchical alignment and high-frequency components are optimal for contrastive uniformity. HyperDepth can thus be interpreted as searching for such an equilibrium rather than enforcing a fixed trade-off.

- **The hierarchical energy preserves semantic depth.** The prototype result (**Proposition 1**) shows that the energy-based alignment learns a tree of prototypes in which leaf nodes align with fine-grained cluster means and internal nodes align with non-negative mixtures of descendant leaf means. This explains why the learned prototype tree reflects nested semantic structure instead of collapsing to a flat geometry, complementing the qualitative behavior visualized in Figure 4.

## C. Details for Algorithm Implementation

The overall training procedure of HyperDepth is summarized in Algorithm 1, and the top-down greedy path inference used in the global alignment loss is given in Algorithm 2.

---

**Algorithm 1** Training Procedures for HyperDepth

---

**Require:** Hypergraph $\mathcal{H} = (\mathcal{V}, \mathcal{E})$, node features $\mathbf{X}$, hyperparameters $(R, K, \tau_{\text{nce}}, \tau_{\text{proto}}, \lambda_{\text{coh}}, \gamma, T_{\max}, T_{\text{rect}})$

1: Initialize encoder parameters $\Phi$, projection heads $g_{\text{loc}}, g_{\text{sem}}$, and prototypes $\mathcal{C} = \{\mathcal{C}^{(k)}\}_{k=1}^{K}$
2: **for** epoch $= 1$ to $T_{\max}$ **do**
3:     *// 1. Dual-view augmentation*
4:     Sample two augmented views by applying feature masking and Gaussian noise:
5:     $\tilde{\mathbf{X}}^{(v)} \leftarrow \mathcal{T}_{\text{feat}}^{(v)}(\mathbf{X}), \quad v \in \{1, 2\}$ {Hypergraph Laplacian $\mathbf{L}$ remains fixed}
6:     *// 2. Decoupled spectral encoding with gating*
7:     **for** $v \in \{1, 2\}$ **do**
8:         $\mathbf{H}^{(v)} \leftarrow f_{\Phi}(\tilde{\mathbf{X}}^{(v)}; \mathbf{L}, R)$ {node embeddings $h_i^{(v)}$}
9:     **end for**
10:    *// 3. Optional geometric rectification of prototypes*
11:    **if** epoch mod $T_{\text{rect}} = 0$ **then**
12:       Compute semantic embeddings $z_i^{\text{sem}} = g_{\text{sem}}(h_i^{(1)})$
13:       Update $\mathcal{C}$ by running level-wise clustering on $\{z_i^{\text{sem}}\}_{i \in \mathcal{V}}$ (prototypes updated off-graph)
14:    **end if**
15:    *// 4. Local contrastive loss*
16:    **for** $v \in \{1, 2\}$ **do**
17:       $z_i^{(v)} \leftarrow g_{\text{loc}}(h_i^{(v)})$ for all $i \in \mathcal{V}$
18:    **end for**
19:    Compute local InfoNCE loss $\mathcal{L}_{\text{Local}}$ using $\{z_i^{(1)}, z_i^{(2)}\}$ (defined in Section 5.4)
20:    *// 5. Global energy-based hierarchical alignment*
21:    Compute semantic embeddings $z_i^{\text{sem}} = g_{\text{sem}}(h_i^{(1)})$ for all $i$
22:    **for** each node $i \in \mathcal{V}$ **do**
23:       Infer positive path $\mathcal{P}_i^+$ by top-down greedy search (Alg. 2)
24:       Construct a set of negative paths $\mathbb{N}_{\text{neg}}(i)$ by perturbing one level along $\mathcal{P}_i^+$
25:    **end for**
26:    Compute global energy-based loss $\mathcal{L}_{\text{Global}}$ using $z_i^{\text{sem}}$, $\mathcal{P}_i^+$ and $\mathbb{N}_{\text{neg}}(i)$ (defined in Section 5.3)
27:    *// 6. Joint optimization*
28:    $\mathcal{L}_{\text{Total}} = \mathcal{L}_{\text{Local}} + \gamma \mathcal{L}_{\text{Global}}$
29:    Update $\Phi$, $g_{\text{loc}}$, $g_{\text{sem}}$ and $\mathcal{C}$ by backpropagation and AdamW
30: **end for**

---

**Algorithm 2** Top-Down Greedy Path Inference for Node $i$

---

**Require:** Semantic embedding $z_i^{\text{sem}}$, prototype sets $\{\mathcal{C}^{(k)}\}_{k=1}^{K}$

1: **Level 1 (root):**
2: $\hat{c}_i^{(1)} = \arg\max_{c \in \mathcal{C}^{(1)}} \langle z_i^{\text{sem}}, c \rangle$
3: **for** $k = 2$ to $K$ **do**
4:    Let $\mathcal{C}^{(k)}(\hat{c}_i^{(k-1)})$ be the children of $\hat{c}_i^{(k-1)}$
5:    Select child maximizing node–prototype affinity plus coherence:

$$\hat{c}_i^{(k)} = \arg \max_{c \in \mathcal{C}^{(k)}(\hat{c}_i^{(k-1)})} \left( \langle z_i^{\text{sem}}, c \rangle + \lambda_{\text{coh}} \langle \hat{c}_i^{(k-1)}, c \rangle \right)$$

6: **end for**
7: **Return** positive path $\mathcal{P}_i^+ = (\hat{c}_i^{(1)}, \ldots, \hat{c}_i^{(K)})$

---

**Path perturbation and complexity.** For the global loss in Section 5.3, we treat $\mathcal{P}_i^+$ produced by Algorithm 2 as the positive semantic path of node $i$. Negative paths $\tilde{\mathcal{P}} \in \mathbb{N}_{\text{neg}}(i)$ are generated by *path perturbation*: we sample a level $k \in \{1, \ldots, K\}$ and replace $\hat{c}_i^{(k)}$ with a different prototype in the same level, while keeping all other levels unchanged. This creates hard negatives that share most of the hierarchical context but violate the semantic structure at one level. Because the search at each level is restricted to the children of the previously selected prototype, the greedy procedure avoids enumerating all leaf nodes or full paths, reducing the search complexity from exponential in $K$ to $O(K\bar{B})$, where $\bar{B}$ is the average branching factor of the prototype tree.

# D. Further Details for Experimental Studies

In this section, we provide additional details of our experimental setup to facilitate reproducibility. We report dataset statistics, implementation specifics of HyperDepth, and the exact hyperparameter configurations used for each dataset.

## D.1. Dataset Statistics

We evaluate HyperDepth on 15 real-world hypergraph datasets that cover both homophilic and heterophilic regimes. The upper block of Table 4 lists the homophilic benchmarks, while the lower block lists the heterophilic datasets. For each dataset, we report the number of nodes, hyperedges, classes, and input feature dimensions.

*Table 4.* Statistics of the 15 datasets. The *upper* block reports homophilic datasets while the *lower* block reports heterophilic datasets.

| Homophilic | Cora-C | Cora-A | Citeseer | Pubmed | 20News | ModelNet40 | NTU2012 | Mushroom | Zoo |
|---|---|---|---|---|---|---|---|---|---|
| Nodes | 1,434 | 2,388 | 1,458 | 3,840 | 16,242 | 12,311 | 2,012 | 8,124 | 101 |
| Edges | 1,579 | 1,072 | 1,079 | 7,963 | 100 | 12,311 | 2,012 | 298 | 43 |
| Classes | 7 | 7 | 6 | 3 | 4 | 40 | 67 | 2 | 7 |
| Features | 1,433 | 1,433 | 3,703 | 500 | 100 | 100 | 100 | 22 | 16 |

| Heterophilic | Actor | Amazon | Twitch | Pokec | House | Senate |
|---|---|---|---|---|---|---|
| Nodes | 16,255 | 22,299 | 16,812 | 14,998 | 1,290 | 282 |
| Edges | 10,164 | 2,090 | 2,627 | 2,406 | 1,630 | 597 |
| Classes | 3 | 5 | 2 | 2 | 2 | 2 |
| Features | 50 | 111 | 7 | 65 | 2 | 2 |

## D.2. Further Details for Implementation

**Evaluation protocol.** We strictly follow the standard *linear evaluation* protocol adopted in prior work such as HyFi (Roh et al., 2024). Training proceeds in two stages: (1) **Unsupervised pre-training**: The encoder is trained on the full hypergraph (structure and features) without using labels, and is optimized only with the proposed objective $\mathcal{L}_{\text{Total}}$; (2) **Linear evaluation:** After pre-training, we *freeze* the encoder and use it to generate node embeddings. A multinomial logistic regression classifier is then trained on these fixed representations. For all datasets, we randomly split nodes into training/validation/test sets with a ratio of 10% / 10% / 80%, and report mean accuracy $\pm$ standard deviation over 5 runs with different random seeds.

**Model architecture and optimization.** We adopt ChebNetII (He et al., 2022) as the backbone encoder. The Chebyshev polynomial order $R$ is selected on the validation split (with the final dataset-specific values reported in Table 5). The hidden dimension $d$ is chosen between 128 and 512, depending on the dataset scale. We optimize the model using AdamW with an initial learning rate generally set to $1 \times 10^{-3}$, a weight decay ranging from $1 \times 10^{-4}$ to $5 \times 10^{-4}$, and a cosine-annealing learning-rate scheduler.

**Geometric rectification of prototypes.** Although the downstream task is node classification, the quality of the latent semantic structure is crucial for representation learning. In practice, randomly initialized prototypes may drift towards low-density regions in the non-convex contrastive landscape, which can trigger collapse. To prevent this, we apply a *geometric rectification* step that periodically re-anchors prototypes to empirical density peaks of the current embedding distribution. This can be viewed as a hard EM–style update for the energy function: given fixed embeddings, we reassign nodes to their nearest prototypes and move each prototype to the (normalized) mean of its assigned nodes. Rectification is performed off-graph with gradients on $\mathcal{C}$ frozen, and is invoked every $T_{\text{rect}} = 5$ epochs in our implementation. The procedure is summarized in Algorithm 3.

---

**Algorithm 3** Geometric rectification (iterative prototype re-anchoring)

---

1: **Input:** node embeddings $\mathbf{H}$, prototype hierarchy $\mathcal{C} = \{\mathcal{C}^{(1)}, \ldots, \mathcal{C}^{(K)}\}$
2: **if** current_epoch $\% \, T_{\text{rect}} == 0$ **and** current_epoch $> 0$ **then**
3:     Detach $\mathbf{H}$ from the computation graph;
4:     **for** each level $k = 1, \ldots, K$ **do**
5:         *// Assignment: nearest-prototype clustering*
6:         $z_i^{(k)} \leftarrow \arg\min_j \|h_i - c_j^{(k)}\|_2$ for all nodes $i$;
7:         *// Update: move prototypes to geometric centers*
8:         **for** each prototype $c_j^{(k)} \in \mathcal{C}^{(k)}$ **do**
9:             $c_j^{(k)} \leftarrow \frac{1}{|\{i : z_i^{(k)} = j\}|} \sum_{i : z_i^{(k)} = j} h_i$;
10:            Normalize to the unit hypersphere:
11:             $c_j^{(k)} \leftarrow c_j^{(k)} / \|c_j^{(k)}\|_2$;
12:         **end for**
13:     **end for**
14: **end if**

---

### D.3. Hyperparameter Settings

We implement HyperDepth in PyTorch and optimize all parameters with AdamW. For each dataset, we perform a grid search on the validation split within a shared search space. Concretely, the number of neighbors $k$ is chosen from $\{2, 5, 10, 15, 20\}$, the prototype temperature $\tau_{\text{proto}}$ from $\{0.01, 0.05, 0.1, 0.2, 0.5\}$, and the balance coefficient $\gamma$ from $\{0.1, 0.5, 1.0, 2.0, 5.0\}$. The hidden dimension $d$ is selected from $\{128, 256, 512, 1024\}$, and the number of hierarchy levels $K$ from $\{1, 2, 3, 4\}$.

For all experiments we use the same search ranges; only the best configuration on the validation set is reported. The final hyperparameter choices for each of the 15 datasets are summarized in Table 5 to facilitate reproducibility.

*Table 5.* Detailed hyperparameter settings for all 15 benchmark datasets.

| Dataset | Hyperparameter Setting | |
|---|---|---|
| **Cora-C** | Learning Rate: $1 \times 10^{-3}$
Weight Decay: $1 \times 10^{-4}$
Hidden Size ($d$): 256
Poly. Order ($R$): 8 | Levels ($K$): 3
Structure: [16, 32, 64]
Balance $\gamma$: 1.0
Neighbors ($k$): 5     Temp. $\tau$: 0.2 |
| **Cora-A** | Learning Rate: $1 \times 10^{-3}$
Weight Decay: $5 \times 10^{-4}$
Hidden Size ($d$): 128
Poly. Order ($R$): 4 | Levels ($K$): 2
Structure: [16, 32]
Balance $\gamma$: 0.5
Neighbors ($k$): 5     Temp. $\tau$: 0.2 |
| **Citeseer** | Learning Rate: $1 \times 10^{-3}$
Weight Decay: $5 \times 10^{-4}$
Hidden Size ($d$): 512
Poly. Order ($R$): 8 | Levels ($K$): 3
Structure: [16, 32, 64]
Balance $\gamma$: 1.0
Neighbors ($k$): 5     Temp. $\tau$: 0.1 |
| **Pubmed** | Learning Rate: $1 \times 10^{-3}$
Weight Decay: $3 \times 10^{-4}$
Hidden Size ($d$): 256
Poly. Order ($R$): 8 | Levels ($K$): 4
Structure: [16, 32, 64, 128]
Balance $\gamma$: 1.0
Neighbors ($k$): 5     Temp. $\tau$: 0.2 |
| **20News** | Learning Rate: $1 \times 10^{-3}$
Weight Decay: $5 \times 10^{-4}$
Hidden Size ($d$): 512
Poly. Order ($R$): 10 | Levels ($K$): 2
Structure: [20, 64]
Balance $\gamma$: 1.0
Neighbors ($k$): 10     Temp. $\tau$: 0.1 |

**Table 5 –** *continued from previous page*

| Dataset | Hyperparameter Setting | |
|---------|------------------------|---|
| **ModelNet40** | Learning Rate: $5 \times 10^{-3}$
Weight Decay: $5 \times 10^{-4}$
Hidden Size $(d)$: 512
Poly. Order $(R)$: 7 | Levels $(K)$: 3
Structure: [40, 64, 128]
Balance $\gamma$: 1.0
Neighbors $(k)$: 10    Temp. $\tau$: 0.1 |
| **NTU2012** | Learning Rate: $1 \times 10^{-3}$
Weight Decay: $5 \times 10^{-4}$
Hidden Size $(d)$: 512
Poly. Order $(R)$: 8 | Levels $(K)$: 4
Structure: [16, 32, 64, 128]
Balance $\gamma$: 1.0
Neighbors $(k)$: 5    Temp. $\tau$: 0.2 |
| **Mushroom** | Learning Rate: $1 \times 10^{-3}$
Weight Decay: $5 \times 10^{-4}$
Hidden Size $(d)$: 256
Poly. Order $(R)$: 4 | Levels $(K)$: 2
Structure: [8, 16]
Balance $\gamma$: 1.0
Neighbors $(k)$: 5    Temp. $\tau$: 0.2 |
| **Zoo** | Learning Rate: $1 \times 10^{-3}$
Weight Decay: $5 \times 10^{-4}$
Hidden Size $(d)$: 128
Poly. Order $(R)$: 4 | Levels $(K)$: 4
Structure: [7, 16, 32, 64]
Balance $\gamma$: 1.0
Neighbors $(k)$: 5    Temp. $\tau$: 0.2 |
| **Actor** | Learning Rate: $1 \times 10^{-3}$
Weight Decay: $5 \times 10^{-4}$
Hidden Size $(d)$: 512
Poly. Order $(R)$: 8 | Levels $(K)$: 2
Structure: [16, 32]
Balance $\gamma$: 2.0
Neighbors $(k)$: 5    Temp. $\tau$: 0.1 |
| **Amazon** | Learning Rate: $2 \times 10^{-3}$
Weight Decay: $5 \times 10^{-4}$
Hidden Size $(d)$: 512
Poly. Order $(R)$: 8 | Levels $(K)$: 2
Structure: [32, 64]
Balance $\gamma$: 1.0
Neighbors $(k)$: 5    Temp. $\tau$: 0.2 |
| **Twitch** | Learning Rate: $1 \times 10^{-3}$
Weight Decay: $5 \times 10^{-4}$
Hidden Size $(d)$: 512
Poly. Order $(R)$: 8 | Levels $(K)$: 4
Structure: [16, 32, 64, 128]
Balance $\gamma$: 1.0
Neighbors $(k)$: 5    Temp. $\tau$: 0.2 |
| **Pokec** | Learning Rate: $1 \times 10^{-3}$
Weight Decay: $5 \times 10^{-4}$
Hidden Size $(d)$: 256
Poly. Order $(R)$: 8 | Levels $(K)$: 3
Structure: [16, 32, 64]
Balance $\gamma$: 1.0
Neighbors $(k)$: 5    Temp. $\tau$: 0.2 |
| **House** | Learning Rate: $1 \times 10^{-3}$
Weight Decay: $5 \times 10^{-4}$
Hidden Size $(d)$: 128
Poly. Order $(R)$: 4 | Levels $(K)$: 2
Structure: [16, 32]
Balance $\gamma$: 0.5
Neighbors $(k)$: 5    Temp. $\tau$: 0.1 |
| **Senate** | Learning Rate: $1 \times 10^{-3}$
Weight Decay: $5 \times 10^{-4}$
Hidden Size $(d)$: 256
Poly. Order $(R)$: 4 | Levels $(K)$: 3
Structure: [16, 32, 64]
Balance $\gamma$: 1.0
Neighbors $(k)$: 5    Temp. $\tau$: 0.2 |

## D.4. Extended Ablation Analysis

Table 6 reports the complete ablation results of HyperDepth on all 15 datasets. We evaluate four variants: (i) **w/o High-pass**, which removes the high-frequency branch and keeps only the low-pass channel; (ii) **w/o Low-pass**, which removes the low-frequency branch and keeps only the high-pass channel; (iii) **w/o** $\mathcal{L}_{\text{Global}}$, which drops the hierarchical alignment loss and trains only with the local contrastive objective; and (iv) **w/o** $\mathcal{L}_{\text{Local}}$, which removes the contrastive branch and optimizes only the global alignment term. The **Full Model** keeps all components.

*Table 6.* Ablation study of HyperDepth on all 15 datasets. We report the mean accuracy (%) $\pm$ standard deviation. The best results are highlighted in **bold**. The *upper* block displays results on homophilic datasets, while the *lower* block shows heterophilic datasets.

| Variant | Cora-C | Citeseer | Pubmed | Cora-A | Zoo | 20News | Mushroom | NTU2012 | ModelNet40 |
|---|---|---|---|---|---|---|---|---|---|
| w/o High-pass | $80.98 \pm 1.21$ | $70.80 \pm 2.10$ | $81.30 \pm 0.91$ | $79.69 \pm 0.69$ | $85.31 \pm 3.55$ | $78.79 \pm 0.66$ | $97.19 \pm 0.73$ | $78.61 \pm 1.38$ | $95.44 \pm 0.12$ |
| w/o Low-pass | $80.30 \pm 1.31$ | $70.54 \pm 1.27$ | $72.18 \pm 1.31$ | $68.62 \pm 1.70$ | $78.77 \pm 4.27$ | $74.85 \pm 0.71$ | $96.33 \pm 0.68$ | $73.51 \pm 0.88$ | $93.25 \pm 0.67$ |
| w/o $\mathcal{L}_{\text{Global}}$ | $80.50 \pm 0.99$ | $69.97 \pm 1.74$ | $80.38 \pm 1.21$ | $79.21 \pm 0.85$ | $84.81 \pm 4.24$ | $78.63 \pm 0.61$ | $98.31 \pm 0.43$ | $78.45 \pm 1.32$ | $94.27 \pm 0.48$ |
| w/o $\mathcal{L}_{\text{Local}}$ | $78.93 \pm 1.05$ | $66.97 \pm 1.76$ | $77.18 \pm 0.60$ | $56.10 \pm 1.08$ | $83.58 \pm 4.35$ | $77.45 \pm 0.50$ | $97.37 \pm 0.69$ | $76.05 \pm 1.23$ | $94.05 \pm 0.53$ |
| **Full Model** | $\mathbf{82.69 \pm 1.20}$ | $\mathbf{72.80 \pm 1.40}$ | $\mathbf{83.48 \pm 0.36}$ | $\mathbf{80.71 \pm 1.17}$ | $\mathbf{89.01 \pm 2.62}$ | $\mathbf{80.87 \pm 0.40}$ | $\mathbf{99.71 \pm 0.44}$ | $\mathbf{79.34 \pm 0.82}$ | $\mathbf{97.61 \pm 0.16}$ |

| Variant | Actor | Amazon | Twitch | Pokec | House | Senate |
|---|---|---|---|---|---|---|
| w/o High-pass | $81.29 \pm 0.40$ | $27.52 \pm 0.36$ | $51.46 \pm 0.56$ | $57.04 \pm 0.16$ | $75.38 \pm 1.07$ | $74.47 \pm 2.05$ |
| w/o Low-pass | $80.80 \pm 0.34$ | $26.59 \pm 0.67$ | $51.38 \pm 0.35$ | $57.04 \pm 0.44$ | $75.82 \pm 0.82$ | $\mathbf{77.88 \pm 2.56}$ |
| w/o $\mathcal{L}_{\text{Global}}$ | $80.74 \pm 0.56$ | $26.91 \pm 0.35$ | $51.92 \pm 0.62$ | $57.06 \pm 0.47$ | $75.61 \pm 0.75$ | $75.88 \pm 2.46$ |
| w/o $\mathcal{L}_{\text{Local}}$ | $80.49 \pm 0.35$ | $26.02 \pm 0.42$ | $51.65 \pm 0.22$ | $56.71 \pm 0.40$ | $75.15 \pm 0.12$ | $75.12 \pm 2.25$ |
| **Full Model** | $\mathbf{83.09 \pm 0.21}$ | $\mathbf{28.68 \pm 0.29}$ | $\mathbf{52.03 \pm 0.63}$ | $\mathbf{57.25 \pm 0.50}$ | $\mathbf{76.10 \pm 0.91}$ | $76.19 \pm 1.78$ |

Across homophilic datasets (upper block), the full HyperDepth consistently achieves the best accuracy or ties for the best in almost all cases, and attains the best average rank. On Cora-C, Citeseer, and Pubmed, removing either $\mathcal{L}_{\text{Local}}$ or $\mathcal{L}_{\text{Global}}$ leads to a clear drop compared with the full model, while the two branches remain competitive with each other. This pattern supports **Theorem 1**: local contrastive learning and global hierarchical alignment play complementary roles, and using both objectives enables embeddings that preserve semantic depth while remaining discriminative at the instance level. On datasets like 20News, Mushroom, and NTU2012, the gap between the Full Model and each variant is slightly smaller but still evident, suggesting that HyperDepth does not simply overfit to a particular benchmark but provides a stable gain on diverse homophilic hypergraphs.

On heterophilic datasets (lower block), the effect of each component is also pronounced, but with one exception. The Full Model achieves the highest mean accuracy on Actor, Amazon, Twitch, Pokec, and House. On Senate, **w/o Low-pass** obtains the highest mean accuracy ($77.88 \pm 2.56$), while the Full Model reaches $76.19 \pm 1.78$; however, the larger variance and overlapping standard-deviation ranges suggest that this should be viewed as a dataset-specific exception rather than a stable advantage of removing the low-pass branch. Across the heterophilic block, removing either $\mathcal{L}_{\text{Local}}$ or $\mathcal{L}_{\text{Global}}$ consistently reduces the mean accuracy relative to the Full Model, indicating that instance-level discrimination and global hierarchical guidance remain complementary under heterophily. These observations further support the equilibrium view in Theorem 1, where neither pure uniformity nor pure hierarchy alone is sufficient.

From the spectral side, removing the low-pass branch is usually more harmful than removing the high-pass branch, especially on homophilic benchmarks such as Cora-A, Zoo, and ModelNet40. This agrees with **Theorem 2**, suggesting that low-frequency information often carries the main global semantic component, while high-frequency information provides local refinements. Nevertheless, this trend is not universal: on House the high-pass-only variant (**w/o Low-pass**) is slightly stronger than the low-pass-only variant (**w/o High-pass**), and on Senate it attains the highest mean accuracy but with a larger standard deviation. Thus, the spectral ablation should be interpreted as evidence that the two spectral channels are complementary, rather than as evidence that one channel is uniformly superior. The performance gaps between **w/o High-pass** and the Full Model on most datasets further indicate that high-frequency information is not redundant; when combined with low-pass information through the adaptive gating mechanism, it improves local discrimination.

Overall, these results underscore the effectiveness of HyperDepth as a unified framework for hypergraph contrastive learning. By jointly leveraging decoupled spectral channels and a hierarchical prototype tree, HyperDepth achieves the best mean accuracy among all ablation variants on 14 out of 15 datasets and remains competitive on Senate, where the high-pass-only variant obtains a higher but more variable mean. Moreover, the Full Model consistently improves over the two objective-level ablations, w/o $\mathcal{L}_{\text{Global}}$ and w/o $\mathcal{L}_{\text{Local}}$, supporting the complementarity between local contrastive learning and global hierarchical alignment. These results suggest that the full spectral–hierarchical design is generally beneficial and robust, without implying that every component dominates on every dataset.

### D.5. Hyperparameter Sensitivity Analysis

In this part, we present a comprehensive hyperparameter sensitivity analysis. We evaluate HyperDepth on 15 datasets while varying five key hyperparameters: the balance coefficient $\gamma$, the hidden dimension $d$, the order of Chebyshev polynomial approximation $R$, the number of hierarchy levels in the prototype tree ($K$), and the prototype temperature $\tau_{\text{proto}}$. Across all settings, the trends are consistent with the main-text results in Figure 6. For each dataset, performance varies only mildly as we sweep these hyperparameters over their ranges. Most datasets achieve near-optimal accuracy within a broad neighbourhood of the chosen defaults, indicating that HyperDepth does not require delicate tuning and is robust across both homophilic and heterophilic hypergraphs.

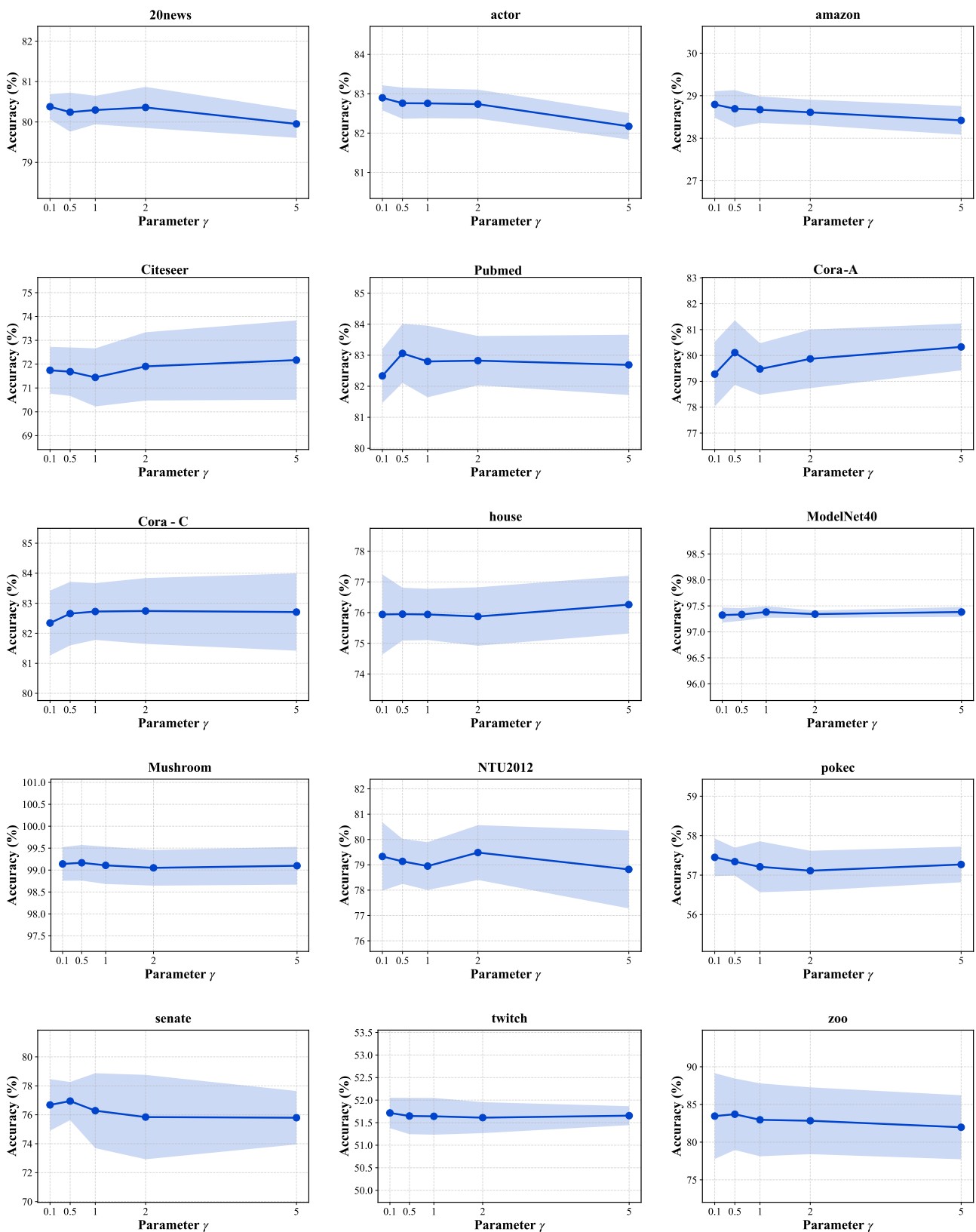

*Figure 7.* Sensitivity analysis for the balance coefficient $\gamma$ across all 15 datasets.

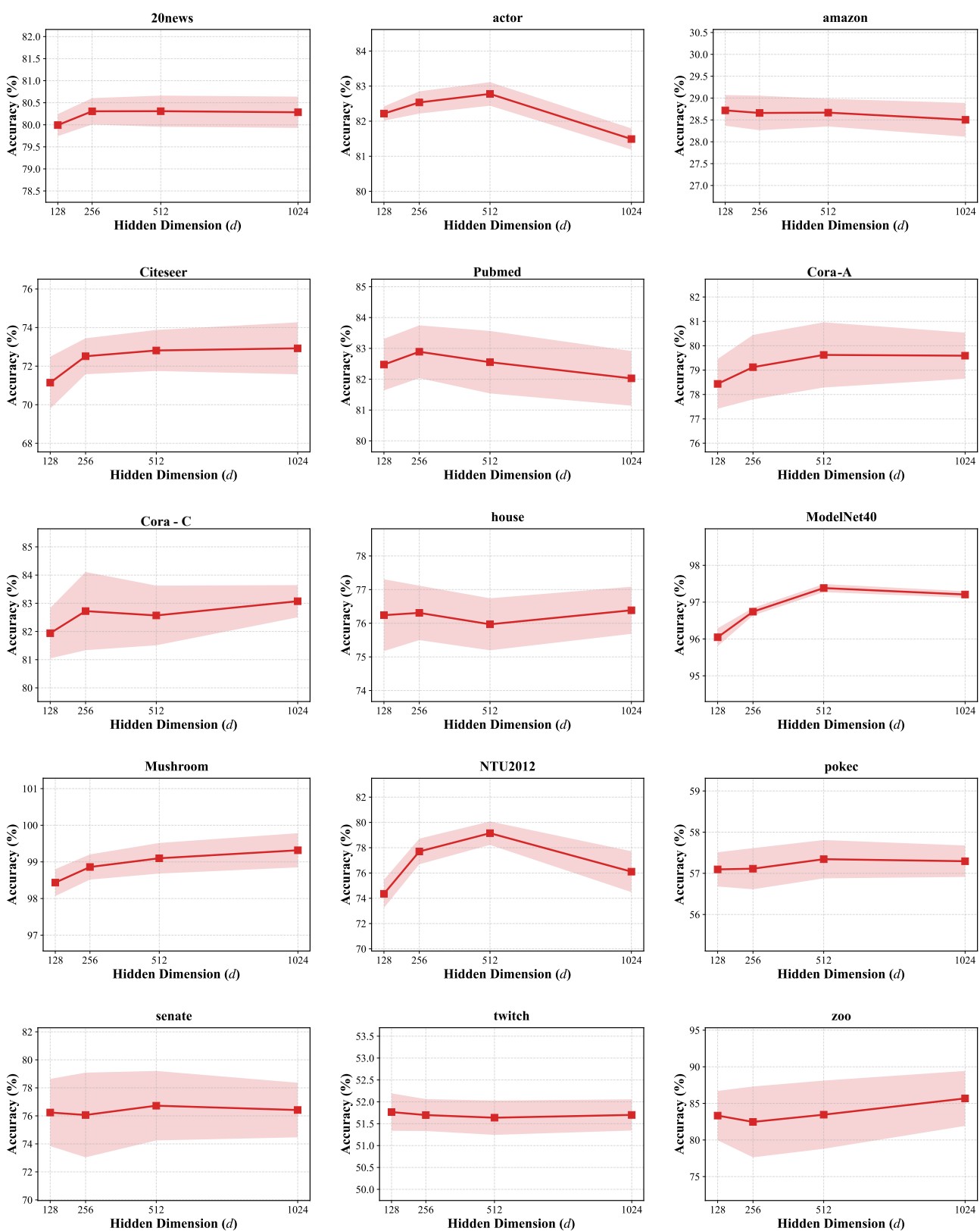

*Figure 8.* Sensitivity analysis for the hidden dimension $d$ across all 15 datasets.

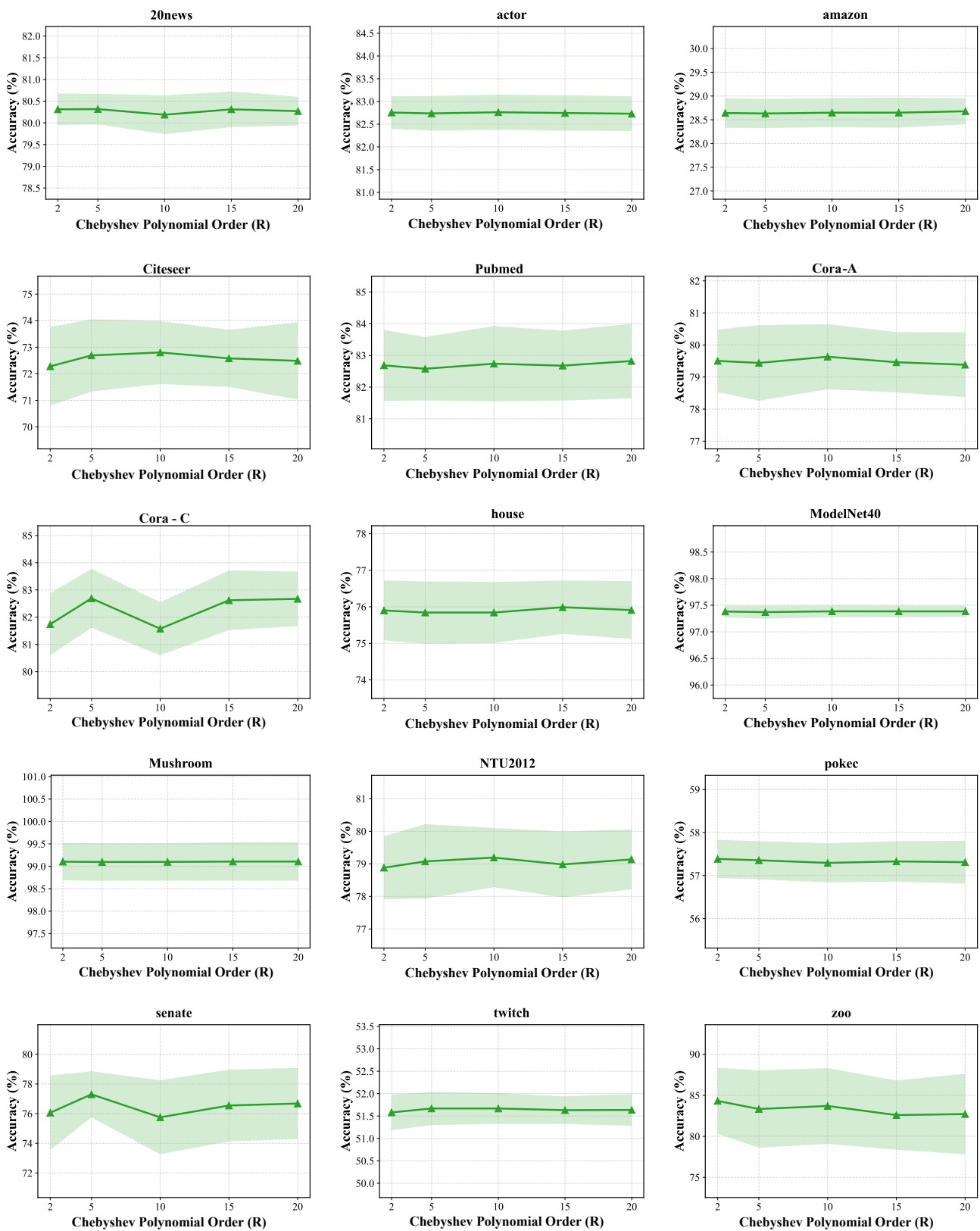

*Figure 9.* Sensitivity analysis for the order of Chebyshev polynomial approximation $R$ across all 15 datasets.

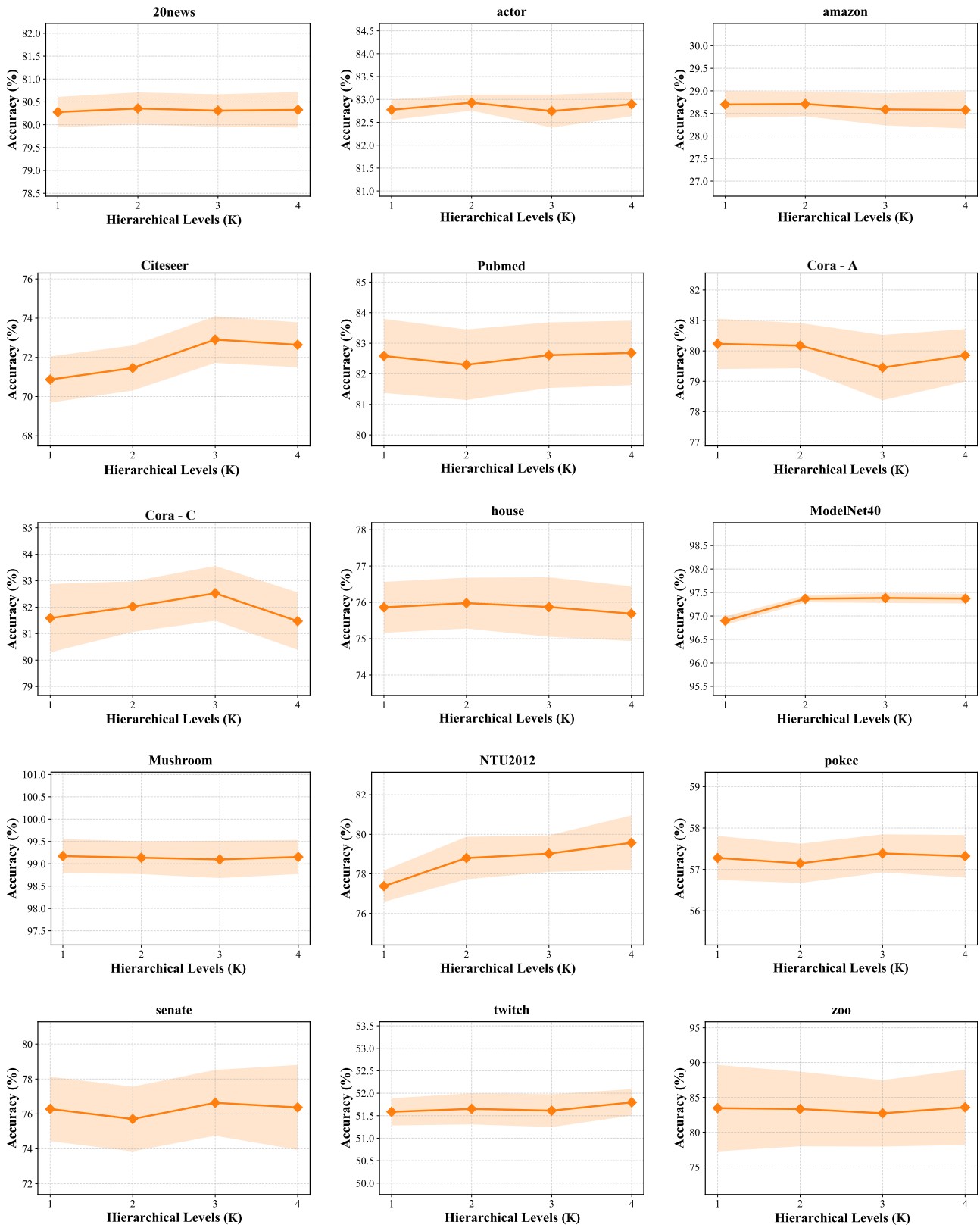

*Figure 10.* Sensitivity analysis for the number of hierarchy levels in the prototype tree ($K$) across all 15 datasets.

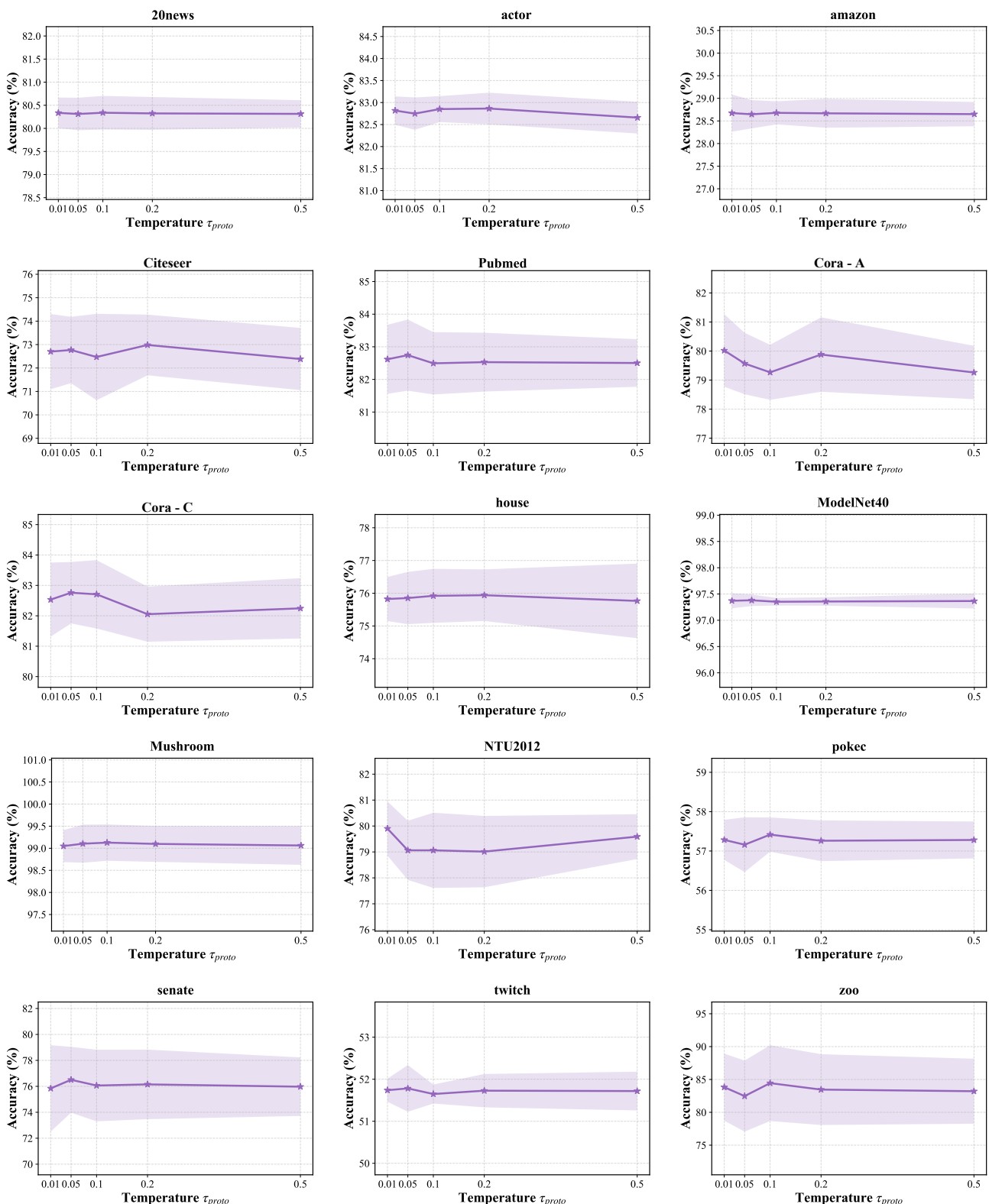

*Figure 11.* Sensitivity analysis for the prototype temperature $\tau_{\text{proto}}$ across all 15 datasets.

