# OpenReview forum: "Towards Hierarchy–Uniformity Equilibrium: Recovering Semantic Depth in Hypergraph Contrastive Learning"
_ICML.cc/2026/Conference — ICML 2026 spotlight_

### Official Review · Reviewer_dDay · 2026-02-27

**Soundness:** 3
**Presentation:** 4
**Significance:** 3
**Originality:** 3
**Overall Recommendation:** 5
**Confidence:** 5

**Summary:**

Focusing on hypergraph contrastive learning, the authors introduce a meaningful problem called Hierarchy-Uniformity Conflict (HUC), whose geometric manifestation is semantic flattening, where the semantic depth of hyperedges collapses into a nearly flat cloud of instances. For problem-solving, they propose a hypergraph contrastive learning framework, HyperDepth, that moves representations toward a hierarchy-uniformity equilibrium by jointly coordinating spectral and geometric signals. Technically, viewing hypergraph contrastive learning through the lens of HUC is, in my opinion, a fresh and interesting perspective that is worth further investigation. The authors also provide theoretical insights showing that the local contrastive and global hierarchical objectives operate on orthogonal spectral components and admit equilibrium embeddings that preserve semantic depth while retaining instance-level discrimination. Extensive experiments on both homophilic and heterophilic datasets show that HyperDepth achieves strong performance compared with several baselines.

**Compliance With Llm Reviewing Policy:**

Affirmed.

**Final Justification:**

The authors have solved all my concerns, and I hope the authors will incorporate these clarifications in the revision.

**Key Questions For Authors:**

1. Why do you choose an energy-based NCE objective on node-path pairs, as illustrated in Figure 3?

2. The experimental studies cover both homophilic and heterophilic hypergraph benchmark datasets, which is comprehensive. However, I am curious whether HyperDepth benefits more on homophilic datasets or on heterophilic ones.

3. Figure 5 intuitively illustrates the learned embeddings and hierarchical prototypes and shows some interesting patterns. Could you clarify which dataset this visualization is based on, and whether other datasets exhibit similar behavior? Related to the Weakness points that I am concerned, how should we interpret this visualization in terms of the unified roles of the global hierarchical alignment module (w.r.t $L_{Global}$) and local-discriminative contrastive module (w.r.t $L_{Local}$)?

**Limitations:**

yes

**Strengths And Weaknesses:**

**Strengths**
1. The motivation and introduction of the HUC problem are interesting.

2. The proposed HyperDepth framework is novel, combining a decoupled spectral encoder and adaptive gating so that high-frequency components emphasize local instance discrimination, while low-frequency components model global hierarchical structure.

3. Both the theoretical analysis and empirical studies are generally convincing.

4. The appendix provides further supportive details.

**Weaknesses**
1. The rationale for why both low- and high-pass components benefit hypergraph contrastive learning could be clarified further.

2. Some mathematical notations need a more careful check to ensure they are well defined and consistent throughout the main text and Appendix.

3. The roles and interaction of the global hierarchical alignment module and the local-discriminative contrastive module would benefit from additional remarks to enhance technical clarity.

---

> ### Author Rebuttal · Authors · 2026-03-29
>
> Thank you for the encouraging review and for highlighting the freshness of the hierarchy--uniformity conflict perspective. We agree that several targeted clarifications would make the paper easier to parse.
>
> - **Why both low- and high-pass components are useful.** The low-pass branch captures **smooth global semantics** and supports coarse-to-fine prototype alignment, while the high-pass branch captures **local deviations and instance uniqueness**, which are essential for contrastive discrimination. HyperDepth needs both because the hierarchy--uniformity conflict is precisely the tension between these two tendencies.
>
> - **Roles and interaction of the two modules.** The global hierarchical alignment module pulls nodes toward coherent prototype paths and restores **semantic depth** in the representation/semantic space. The local contrastive module applies InfoNCE in a separate projection space and preserves **node-level discrimination / uniformity**. They share the same encoder, but act on different spaces and complementary spectral components, which is exactly why they work well together.
>
> - **Notation.** We appreciate this catch and will do a careful notation pass in both the main text and the Appendix to ensure all symbols are consistently defined and reused.
>
> - **Q1 (Why energy-based NCE on node--path pairs?).** A single prototype is not sufficient to encode semantic depth; what matters is a **structured root-to-leaf path**. The energy-based NCE objective is natural here because it makes the correct path a low-energy configuration and one-level-perturbed paths hard negatives. This yields informative near-miss comparisons and sharpens boundaries between closely related hierarchies.
>
> - **Q2 (Homophilic vs. heterophilic).** HyperDepth performs strongly in both regimes, but the relative benefit is particularly visible on heterophilic datasets, where pure neighborhood smoothing is less reliable. In that case, the high-frequency/local branch preserves informative deviations, while the global prototype alignment still provides stable semantic organization beyond noisy local neighborhoods.
>
> - **Q3 (Fig. 5 dataset and interpretation).** The current draft omitted the dataset name in the caption; we will add it explicitly (**Cora-C**). The intended interpretation is: panel (1) = hierarchy without sufficient discrimination, panel (2) = discrimination without a coherent multi-level prototype hierarchy, and panel (3) = their combination under the full objective. Even beyond this plot, the same qualitative tendency is consistent with the ablation results across the 15 benchmarks: removing either branch consistently hurts performance.
>
> We thank the reviewer again for the positive feedback and will incorporate these clarifications in the revision.

---

> > ### Author Rebuttal · Reviewer_dDay · 2026-04-03
> >
> > Thanks for the clarifications from the authors. I will maintain my positive score.

---

### Official Review · Reviewer_jpgx · 2026-02-28

**Soundness:** 4
**Presentation:** 3
**Significance:** 3
**Originality:** 3
**Overall Recommendation:** 6
**Confidence:** 5

**Summary:**

This paper introduces the concept of hierarchy-uniformity conflict, a new problem in the context of hypergraph contrastive learning. HyperDepth model is developed using a decoupled spectral encoding scheme with adaptive gating: high-frequency components focus on local instance discrimination, while low-frequency components capture global hierarchical structure. An energy-based hierarchical alignment module attaches a learnable prototype tree to the representation space and minimizes an energy functional to recover the semantic depth of hyperedges. From a theoretical perspective, under certain assumptions, the authors provide two theorems and one proposition to support the effectiveness of the method. Experiments on 15 hypergraph datasets and 17 supervised and self-supervised baselines show that HyperDepth achieves better performance than all the considered baselines.

**Compliance With Llm Reviewing Policy:**

Affirmed.

**Final Justification:**

Overall, consistent with my original evaluation and further strengthened by the authors' clear rebuttal and additional explanations, I find the novelty, clarity, significance, and technical soundness of this work to be clearly above the acceptance bar for ICML. My final recommendation is an SA.

**Key Questions For Authors:**

(1) Why does Table 2 report only the results of self-supervised methods on heterophilic datasets? Do homophilic datasets and supervised baselines show a similar performance trend?
(2) Although the schematic of HyperDepth is clear and the implementation seems straightforward to reproduce, could you provide more discussion (probably in the Appendix part) on the computational complexity?
(3) Based on the main results in Table 1 and Table 2, can we say that HyperDepth tends to perform more favorably on heterophilic datasets than on homophilic ones? If so, what is your intuition or explanation for this behavior?

**Limitations:**

Yes

**Strengths And Weaknesses:**

Strengths:

(1) In terms of **presentation**, the paper is well-structured; the introduction clearly motivates the problem and the overall readability is above the bar of acceptance.

(2) For **soundness**, the authors provide both theoretical results and extensive empirical evidence, which together suggest that the HyperDepth framework is technically correct, theoretically grounded, and methodologically appropriate.

(3) Regarding **significance**, this submission introduces the new issue of hierarchy-uniformity conflict, which may deepen the understanding of hypergraph contrastive learning (HCL); furthermore, the idea of moving representations toward a hierarchy-uniformity equilibrium by jointly coordinating spectral and geometric signals appears practically useful for the hypergraph learning community.

(4) In terms of **originality**, the authors offer some new insights into HCL, and HyperDepth itself is a novel method equipped with some theoretical results on its key properties

Weaknesses:

(1) Since the theoretical analysis relies on certain assumptions, authors need to elaborate more on the scope and practical plausibility of these assumptions and how flexible they are in the hypergraph learning setting.

(2) Although both the local and global channels contribute to HyperDepth, the key technical differences between the two modules, and how they interact, are not fully clear. I think additional clarification or brief remarks highlighting these key points are needed.

(3) In the experimental section, it is not clearly explained how the linear evaluation protocol differs from the unsupervised pre-training evaluation setup.

---

> ### Author Rebuttal · Authors · 2026-03-29
>
> Thank you for the positive assessment of the novelty, technical soundness, and significance of the work. We especially appreciate the request to sharpen the scope of the theory and the evaluation protocol.
>
> - **Scope and plausibility of the assumptions.** We agree that this deserves clearer explanation. **Assumption 1** is an analysis lens rather than a hard architectural constraint: in practice, HyperDepth only **approximates** this regime through dual-band filtering, separate projection heads, and adaptive gating. **Assumption 2** is meant to be mild: it only requires that coarse semantics admit smoother low-frequency structure; it does **not** assume a perfectly labeled or perfectly known hierarchy. We will clarify that the theory characterizes the regime HyperDepth targets, rather than claiming an exact description of every hypergraph.
>
> - **Difference and interaction of the two modules.** The **global** module performs **node--path matching** in the representation/semantic space and learns a prototype tree that preserves semantic depth. The **local** module performs **node--node matching** across augmented views in a contrastive projection space and enforces instance discrimination / uniformity. Their interaction happens through the shared spectrally enriched encoder, while Theorem 1 explains why they play complementary rather than redundant roles.
>
> - **Linear evaluation vs. unsupervised pre-training.** We will clarify in the main text that the pipeline strictly follows a two-stage protocol: **(1)** unsupervised pre-training on the full hypergraph using $L_{\text{Total}}$ without labels; **(2)** linear evaluation, where the encoder is frozen and a linear classifier is trained to assess representation quality. The goal is to measure the quality of the learned representations, rather than end-to-end supervised fitting performance.
>
> - **Q1 (Why Table 2 only self-supervised methods on heterophilic datasets?).** Our intention in Table 2 was to isolate the comparison **within the self-supervised hypergraph contrastive family** in the most challenging heterophilic regime. On the homophilic side, Table 1 already shows that HyperDepth remains highly competitive even against supervised baselines and achieves the best average rank overall. We will state this scoping choice more explicitly.
>
> - **Q2 (Computational complexity).** We will add a short Appendix summary. Dual-band filtering is implemented with Chebyshev approximation and scales as **$O(R|E|)$**. Positive-path inference avoids exponential path enumeration and is reduced to **$O(\text{depth} \times \bar B)$** by the top-down greedy search, where $\bar B$ is the average branching factor. Prototype rectification is only a periodic off-graph stabilization step. Thus, the added overhead is modest and mainly comes from prototype alignment, not from combinatorial path search.
>
> - **Q3 (Homophily vs. heterophily).** The method is strong in both regimes, but the **relative gain** is indeed somewhat more pronounced on heterophilic benchmarks within the self-supervised family. Our intuition is that heterophily makes pure smoothing less reliable, while informative local deviations become more important. HyperDepth is designed exactly for this setting: it preserves such high-frequency/local cues while still maintaining broader semantic organization through the low-frequency/prototype branch.
>
> We appreciate these suggestions and will incorporate them to make the presentation sharper.

---

> > ### Author Rebuttal · Reviewer_jpgx · 2026-04-02
> >
> > After reading the rebuttal, I am satisfied with the authors' clarifications regarding the scope of the theoretical assumptions, the interaction between the global and local modules, and the evaluation protocol. These responses resolve my main concerns and make the technical contribution of HyperDepth clearer and better motivated. I therefore maintain my positive recommendation for acceptance.
> >
> > Overall, I believe this work is a meaningful contribution of clear significance to the hypergraph learning community and is likely to inspire follow-up research on hierarchical structure in hypergraph contrastive learning.

---

### Official Review · Reviewer_aCVV · 2026-03-04

**Soundness:** 4
**Presentation:** 3
**Significance:** 3
**Originality:** 3
**Overall Recommendation:** 6
**Confidence:** 5

**Summary:**

This submission takes hierarchy-uniformity conflict as a key angle to investigate the semantic-depth collapse of hyperedges under the hypergraph contrastive learning paradigm. The authors propose a framework named HyperDepth, which aims to move representations toward a hierarchy-uniformity equilibrium by jointly coordinating spectral and geometric signals, and it explicitly considers both low- and high-frequency components. They also provide theoretical results supporting the insight that local contrastive and global hierarchical objectives operate on orthogonal spectral components and admit equilibrium embeddings that preserve semantic depth while retaining instance-level discrimination. Experiments show HyperDepth outperforms existing baselines.

**Compliance With Llm Reviewing Policy:**

Affirmed.

**Key Questions For Authors:**

(1) In the full HyperDepth objective, how should we interpret the role of global loss and local loss conceptually? What kind of representation behavior does each term encourage, and how do you decide when the two objectives are “well balanced”?

(2) Are there any specific challenges when dealing with heterophily in hypergraphs, and how do heterophilic neighbors affect the hierarchy–uniformity conflict?

**Limitations:**

Yes

**Strengths And Weaknesses:**

**Strengths:**

(1) The paper introduces a fresh perspective for hypergraph contrastive learning by framing the problem through hierarchy-uniformity conflict. This is a novel perspective, and it is likely to be useful for both researchers and practitioners to reuse or extend in future work.

(2) The paper is clearly written and well structured, and the appendix is fairly comprehensive.

(3) HyperDepth is supported by theoretical analysis and principle-level remarks that help interpret the method design.

(4) The experiment results are generally supportive of the advantages of HyperDepth and suggest good overall soundness.


**Weaknesses:**

(1) While the low- and high-frequency intuition is discussed, the connection to concrete hypergraph structural patterns could be explained more directly (e.g., what kinds of local or global hypergraph patterns are mainly captured by the high-pass vs. low-pass components).

(2) The construction of positive and negative pairs for the local energy-based objective (node-path pairs) is only briefly described. A bit more clarification on the sampling strategy and how trivial or noisy pairs are avoided would help readers better understand the stability of the contrastive learning setup.

(3) A short clarification on what empirical indicators correspond to “flattening” can improve clarity.

---

> ### Author Rebuttal · Authors · 2026-03-29
>
> Thank you for your insightful and encouraging feedback. We appreciate that you view this work as providing a new perspective on hypergraph contrastive learning and recognize our theoretical analysis of the hierarchy–uniformity conflict.
>
> - **Low- vs. high-frequency structural patterns.** We agree this can be made more concrete. In our setting, **low-frequency** components capture patterns that vary smoothly across incident hyperedges, such as coarse communities, repeated co-membership, and shared semantic ancestors. **High-frequency** components capture local deviations from neighborhood averages, such as boundary nodes, rare memberships, or fine-grained distinctions among nodes that appear in otherwise similar hyperedge contexts.
>
> - **Positive/negative construction.** To clarify, the **node-path pairs** belong exclusively to the **global energy-based branch**, while the **local branch** employs standard view-based InfoNCE. For each node $i$, we perform a top-down greedy search (detailed in Appendix D) to identify the positive path $P_i^+$ that maximizes alignment energy. We then generate **hard negatives** by perturbing exactly **one hierarchy level** of this positive path. This specific perturbation creates samples that are structurally similar but semantically conflicting, ensuring that the negative pairs are challenging and non-trivial. In contrast, the **local branch** follows the standard contrastive setup where augmented views of the same node form positive pairs, and all other nodes are treated as negatives.
>
> - **Empirical indicators of semantic flattening.** Empirically, flattening means that the nested coarse-to-fine layout is lost: top/mid/bottom prototypes overlap, within-ancestor distances become close to across-group distances, and embeddings no longer exhibit clear level-specific centroids. This is exactly the contrast we intended between panel (2) and panel (3) of Fig. 5, and we will state it more explicitly.
>
> - **Conceptual role of $L_{\text{Global}}$ and $L_{\text{Local}}$.** $L_{\text{Global}}$ encourages **coarse-to-fine semantic organization** by aligning nodes with prototype paths in the representation/semantic space. $L_{\text{Local}}$ preserves **node identity and contrastive uniformity** in a separate projection space. They are well balanced when stable hierarchical structure and strong instance discrimination coexist; empirically, this typically occurs around a moderate trade-off, and the sensitivity curves are quite stable (often peaking near $\gamma \approx 1$).
>
> - **Heterophily.** Heterophily makes naive smoothing less reliable because nearby nodes may not share labels or fine semantics. In this regime, the hierarchy--uniformity conflict can become sharper: pure smoothing may blur informative differences, while pure uniformity may ignore broader semantic organization. HyperDepth helps precisely because it preserves informative local deviations in the high-frequency/local branch while still recovering broader semantic structure in the low-frequency/prototype branch.
>
> We thank the reviewer again for recommending a strong acceptance and will incorporate these clarifications in the revision.

---

> > ### Author Rebuttal · Reviewer_aCVV · 2026-04-04
> >
> > My concerns have been adequately addressed.

---

### Official Review · Reviewer_nEir · 2026-03-04

**Soundness:** 3
**Presentation:** 3
**Significance:** 3
**Originality:** 3
**Overall Recommendation:** 5
**Confidence:** 4

**Summary:**

In this work, the authors develop a new hypergraph contrastive learning method, called HyperDepth, motivated by the observation that local instance uniqueness and global hierarchical structure are mainly carried by different spectral components: high-frequency variations capture fine-grained distinctions between nearby nodes, whereas low-frequency components encode smooth patterns consistent with coarse semantic groupings. The paper provides a geometric formulation of the tension between instance-level uniformity and hierarchical semantics, introduces the notion of a hierarchy-uniformity equilibrium, and theoretically shows that, under a mild frequency-separation assumption, the local and global objectives act on orthogonal spectral components and admit equilibrium embeddings that preserve both semantic depth and discriminability. To validate the effectiveness and advantage of HyperDepth, the authors conduct experiments on multiple benchmark datasets (including both homophilic and heterophilic ones) and compare against several supervised and self-supervised baselines. The experimental results are generally convincing.

**Compliance With Llm Reviewing Policy:**

Affirmed.

**Final Justification:**

The paper presents a technically solid and novel perspective on hypergraph contrastive learning, and the proposed HyperDepth framework is supported by both theoretical analysis and strong empirical results. After reading the authors' rebuttal, I am satisfied that my main concerns, especially those regarding Algorithm 3, the visualization in Fig. 5, and the interpretation of the spectral components, are largely clarification issues that can be addressed in the final version. Taking both the paper and the rebuttal into account, I remain positive and support acceptance.

**Key Questions For Authors:**

During the rebuttal stage, besides further clarifying the weakness points above, the authors need to consider addressing the following questions:

Q1. Fig. 6 has a typo. Should it be “four” or “three”?

Q2. Is it possible to use other types of high-pass filters in the dual-band spectral filtering module?

Q3. As shown in Fig. 2, why does the global hierarchical alignment module contribute to the top/mid/bottom-level components at the initial/intermediate/final states, respectively?

**Limitations:**

Although the authors provide a short paragraph in the Impact Statement Section that clearly relates this work to broader themes in hypergraph machine learning, it would be helpful to further discuss the potential limitations and how HyperDepth and/or the newly introduced concepts of hierarchy-uniformity conflict and semantic flattening may influence future directions in hypergraph learning (as mentioned in Weakness #4). Considering the page limit, these additional remarks or notes may be included in the Appendix.

**Strengths And Weaknesses:**

Strengths:
1. The concepts of hierarchy-uniformity conflict and semantic flattening are interesting and well defined.
2. HyperDepth is a novel framework for hypergraph contrastive learning.
3. The presentation is good, and the appendix provides additional supportive content.
4. The authors provide both theoretical analysis and experimental studies to demonstrate the soundness of HyperDepth. In particular, the theoretical insights may inspire follow-up work in this direction.

Weaknesses:
1. The role of Algorithm 3 is not clear to me. More clarification on how it works and how it benefits the implementation of HyperDepth is needed.
2. The visualization results shown in Fig. 5 confuse me. Why do subfigures (1) and (2) not reflect the distribution of Red (Top-level), Black (Mid-level), and Green (Bottom-level) points?
3. For clarity, in Section 4.2, why do you use shared weights in the decoupled spectral encoder with the adaptive gating channel?
4. For significance, further directions on how to extend HyperDepth (and/or the concepts of hierarchy-uniformity conflict and semantic flattening) should be discussed in the Appendix.

---

> ### Author Rebuttal · Authors · 2026-03-29
>
> Thank you for the positive assessment and for recognizing the novelty of the hierarchy--uniformity conflict perspective, as well as the strength of the theory and experiments. We believe the remaining concerns are mainly clarification issues and can be directly addressed in the revision.
>
> - **Role of Algorithm 3.** Algorithm 3 is an optional **geometric rectification / prototype re-anchoring** step, rather than an additional objective. It is motivated by Eq. (6): given the current embeddings, we periodically re-center each prototype at the normalized mean of the nodes assigned to it. In practice, this acts like a lightweight hard-EM step that prevents prototype drift toward low-density regions and improves optimization stability.
>
> - **Fig. 5.** We agree that the current caption is too brief. Panels **(1)** and **(2)** illustrate **failure modes**: using only $L_{\text{Global}}$ overly pulls nodes toward prototypes, leading to *prototype collapse*, where distributions become indistinguishable; using only $L_{\text{Local}}$ enforces uniformity, spreading nodes evenly across the manifold, resulting in *semantic flattening* and the loss of clear hierarchical structures. Only  panel **(3)** clearly shows the three-level hierarchy (red/black/green). We will revise the caption and explicitly state the dataset name (**Cora-C**) in the final version.
>
> - **Why shared weights?** The shared weights are used **across the two augmented views**, so both views are embedded into the same latent geometry. This avoids view-specific drift and makes the contrastive comparison meaningful. The separation between local and global roles is then induced by the complementary spectral filters, adaptive gating, and separate heads, rather than by using different encoders.
>
> - **Extensions / limitations.** We agree this would strengthen the paper. We will add a short Appendix discussion on promising extensions, including alternative spectral filters, adaptive hierarchy depth/branching, dynamic hypergraphs, and tasks beyond node classification. We will also explicitly discuss that gains may be smaller when hierarchical semantics are weak or highly noisy.
>
> - **Q1 (Fig. 6 typo).** Yes -- it should be **"three,"** not **"four."**
>
> - **Q2 (other high-pass filters).** Yes. Our framework does not rely on the specific form $g_{\text{high}}(\lambda)=1-e^{-\beta\lambda}$; we chose it because it is smooth, stable, and easy to approximate with Chebyshev polynomials. Other high-pass families can also be used as long as they preserve a meaningful low/high-frequency separation.
>
> - **Q3 (Fig. 2: initial/intermediate/final states).** This figure is an intuitive **coarse-to-fine** illustration rather than a strict stage-wise process. All levels are jointly optimized. Early in training, the model relies more on low-frequency information to establish global semantics; as path assignments refine, high-frequency information increasingly contributes, enhancing instance-level discrimination and making mid- and low-level structures clearer.
>
> We appreciate these suggestions and believe they can all be addressed cleanly in the revision.

---

> > ### Author Rebuttal · Reviewer_nEir · 2026-04-01
> >
> > The authors have addressed all my concerns with clear arguments and explanations. I am therefore happy to raise my score and recommend acceptance.

---

### Decision · Program_Chairs · 2026-04-30

**Decision:**

Accept (spotlight)

**Comment:**

This paper proposes a hypergraph contrastive learning framework that aims to resolve the conflict between hierarchy and uniformity. The method recovers rich semantic information through a decoupled spectral encoder and an energy-based hierarchical alignment module. The superiority of the proposed method is demonstrated by comparing with several baseline methods on 15 datasets and a large number of auxiliary experiments and theoretical analysis. Reviewers generally agreed on the novelty and plausibility of the approach, with main concerns focusing on the clarification of some components, visual explanation, and rationale for design choices, which the authors responded well with additional explanations and analysis in their responses. Overall, the paper is well-motivated, technically solid, and shows superior experimental results in both homophilic and heterophilic scenarios, so it is suitable for acceptance.